# Token Embeddings Violate the Manifold Hypothesis

**Michael Robinson**
Mathematics and Statistics
American University
Washington, DC, USA
michaelr@american.edu

**Sourya Dey**
Galois, Inc.
Arlington, VA, USA
sourya@galois.com

**Tony Chiang**
Department of Mathematics, University of Washington,
Seattle, WA, USA
chiang@math.washington.edu

## Abstract

A full understanding of the behavior of a large language model (LLM) requires our grasp of its input token space. If this space differs from our assumptions, our comprehension of and conclusions about the LLM will likely be flawed. We elucidate the structure of the token embeddings both empirically and theoretically. We present a novel statistical test assuming that the neighborhood around each token has a relatively flat and smooth structure as the null hypothesis. Failing to reject the null is uninformative, but rejecting it at a specific token $\psi$ implies an irregularity in the token subspace in a $\psi$-neighborhood, $B(\psi)$. The structure assumed in the null is a generalization of a manifold with boundary called a *smooth fiber bundle* (which can be split into two spatial regimes – small and large radius), so we denote our new hypothesis test as the "fiber bundle hypothesis." By running our test over several open-source LLMs, each with unique token embeddings, we find that the null is frequently rejected, and so the evidence suggests that the token subspace is not a fiber bundle and hence also not a manifold. As a consequence of our findings, when an LLM is presented with two semantically equivalent prompts, if one prompt contains a token implicated by our test, the response to that prompt will likely exhibit less stability than the other.

## 1 Introduction

A core area for Artificial Intelligence (AI) research is to understand the central components of modern learning systems, e.g. large language models (LLMs), *assuming certain regularity conditions*. Extensive research efforts have been mobilized to tackle the explainability of AI question (Gunning et al. [2019]), leveraging ideas from computer science (Gordon [2024], Wu et al. [2022], Traylor et al. [2021]), information theory (Tan et al. [2024], Wang et al. [2024]), and theoretical physics (Chen et al. [2025], Geshkovski et al. [2023], Lu et al. [2019]), to mathematical insights from high-dimensional probability theory (Huang et al. [2024], Asher et al. [2023]), and algebraic and differential topology (Bradley and Vigneaux [2025], Rathore et al. [2023], Bradley et al. [2022]) either to prove mathematical guarantees or to elucidate model behavior.

In many—if not most—AI research papers, a *manifold hypothesis* is tacitly assumed, that the data are concentrated near a low curvature manifold without boundary. This assumption drives the research just as an assumption of Gaussianity drives classical approaches to data modeling. For instance, the Uniform Manifold Approximation and Projection (UMAP) algorithm (McInnes et al. [2018]) *explicitly* assumes that data are sampled near a low dimensional manifold and cautions the reader that

39th Conference on Neural Information Processing Systems (NeurIPS 2025).

the algorithm may not be reliable if this assumption is violated. Yet many researchers make inferences from UMAP, rarely checking that this assumption holds, and even arguing it is unnecessary. This argument has been routinely rejected in the literature (Jeon et al. [2025], Chari and Pachter [2023]), and so it is important to test this kind of assumption before data modeling. One common testing strategy is simply "plotting the data" to ascertain the empirical distribution. Plotting high-dimensional data, however, is non-trivial, but there are statistical tests to help us understand the underlying distribution (Mann-Whitney, Kolomogorov-Smirnoff, etc.). In this paper, we provide an analogous test for the *regularity* of token embeddings.

The first layer of most LLMs is usually comprised of the token embeddings, which are vector representations of *tokens* (fragments of texts). Thus, a good first step to explain the behavior of an LLM is to understand its initial *token subspace* where these embedding vectors lie. For the remainder of this paper, the term "token"' used in the context of the token subspace will refer to this embedding. Numerous papers (for instance, see Verma et al. [2024], He et al. [2024], Vargas et al. [(to appear], Li et al. [2024], Battle and Gollapudi [2024], Valmeekam et al. [2023], Sclar et al. [2023], Chao et al. [2023]) have pointed to unexpected behaviors exhibited by LLMs that hinge on subtle changes in wording and text layout between apparently similar prompts, suggesting that certain semantically similar tokens may have dramatically different topological neighborhoods in the token subspace. Linguistically, these irregularities may correspond to *polysemy* or *homonyms* (Jakubowski et al. [2020]).

The presence of irregularities imply small changes in language can drive large variation in the token subspace. Moreover, if the token subspace of an LLM is, in fact, *singular*, hence not a manifold, irregularities can persist into the output of the LLM, perhaps unavoidably and potentially even because of its architecture. This is because the input and output of LLMs are a fixed finite set of tokens yet the internal architectures of the model assume a structure resembling a Riemannian manifold: a metrizable space that locally preserves inner products. Not accounting for singularities in the token subspace may thereby confound our understanding of the LLM's behavior.

## 1.1 Contributions

Our contributions are: (1) two novel statistical tests (for manifolds and for fiber bundles; see below), (2) evidence to reject both hypotheses for several LLMs (Section 5.2), and therefore (3) the identification of otherwise unreported structure in the embeddings for these LLMs. We emphasize that while this paper demonstrates the manifold and fiber bundle tests on token embeddings, our *main* contribution is primarily a methodology for hypothesis testing on embedded points.

Because connected manifolds have a unique dimension, and because the dimension of fiber bundles are characterized by Theorem 1, our two tests address the following hypotheses given in Table 1.

Table 1: Hypotheses we test. We assume the token subspace has reach $\tau > 0$ (see definition in Sec. 2), and estimate dimension in the ball centered at token $\psi$ with radius $r < \tau$.

|  | Manifold test | Fiber bundle test |
|---|---|---|
| $H_0$ | There is a unique dimension at $\psi$ | The dimension at $\psi$ in a ball of radius $r$ does not increase as $r$ increases. |
| $H_1$ | There is not a unique dimension at $\psi$ | The dimension at $\psi$ increases at some $r$ |

Both tests are implemented as Algorithm 1. The embedded reach $\tau$ defines an assumption on the curvature of the underlying space. Low reach corresponds to high curvature and *vice versa* (Berenfeld et al. [2022]). Classical Lagrange interpolation asserts that any finite sample of points can be fit exactly by a manifold. The resulting manifold may have extremely high curvature, which is undesirable because it results in numerical instability (Isaacson and Keller [2012]).

To demonstrate our method, Figure 3(a)–(c) shows the application of the manifold test to several synthetic examples. In Section 5.2, we then provide empirical evidence that the token embedding function obtained from each of four different open source LLMs of moderate size—GPT2 (Radford et al. [2019]), Llemma7B (Azerbayev et al. [2024]), Mistral7B (Jiang et al. [2023]), and Pythia6.9B (Biderman et al. [2023])—cannot be a manifold due to non-constant local dimension as well as tokens (points) without an intrinsic dimension. When these tests are rejected at token $\psi$, this implies that $\psi$ is less stable under mathematical transformations than other tokens. In Section 3.2, we show the

endemic nature of these instabilities by proving that *context cannot resolve singularities persistently*. And finally, we provide conditions (Theorem 2) for when singularities can propagate into the output of an LLM for *generic* transformers.

## 1.2  Background and related work

Manifold learning takes its roots in nonlinear dimension reduction of high dimensional data into low dimensional representations of smooth manifolds. While these algorithms have played an important role in modern data science especially in domains such as cosmology (Park et al. [2024]), astrophysics (Makinen et al. [2024], Einig et al. [2023]), dynamical systems (Masuda and Kundu [2022], Parsons and Rogers [2015]), and quantum chemistry (Hemmateenejad et al. [2004]), the main utility of these methods have been for data visualization. For instance, even though UMAP provided an optimal algorithm for fitting sampled data to a manifold, the authors themselves concede that "if the global structure is of primary interest then UMAP may not be the best choice for dimensionality reduction". A manifold learning algorithm will find *some* smooth structure to fit the data rather than "error out" when the data is not sampled from a manifold as it is not a hypothesis test. Grigsby et al. [2023] showed the dimension of ReLU-Nets are inhomogenous, and so when algorithms fit a manifold to data optimally, for instance, the found manifold may have extremely high curvature or dimension. Both of these are undesirable properties in a model, and both implicitly suggest that a manifold would be a poor choice for a representation (Isaacson and Keller [2012]).

To address this concern, Fefferman et al. [2016] presented a test for whether data lie on a manifold with a specified embedded reach, but did not present the values of constants that are critical for practical implementation. More recently, Lim et al. [2023], Von Rohrscheidt and Rieck [2023] present tests for the manifold hypothesis, with promising results on low dimensional datasets. Our paper remedies both issues: we construct a practical implementation of hypothesis tests for manifolds and fiber bundles with a specified embedded reach.

The token input embedding matrix defines the salient subspace of the latent space for an LLM, by specifying where the tokens are located. There is no *a priori* reason to suspect it is a manifold. It has already been shown that local neighborhoods of each token have salient topological structure (Rathore et al. [2023]), and one of the most basic parameters is the *dimension* near any given token in this space. Higher dimension at a token $\psi$ means that $\psi$ has greater density while lower dimensional tokens have less density (Jakubowski et al. [2020]). Intuitively, a densely populated neighborhood $B(\psi)$ will generically occupy a higher dimension than otherwise.

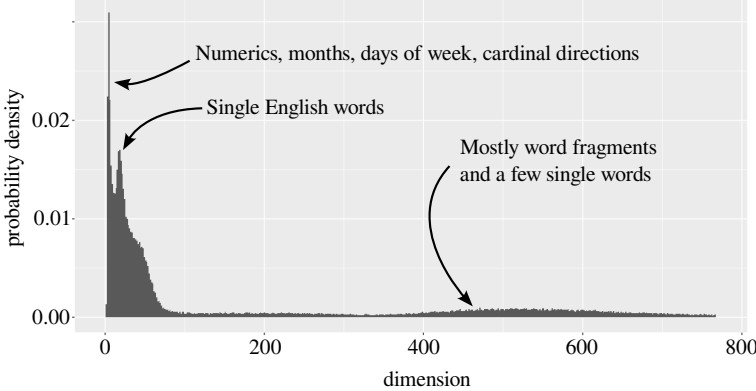

Figure 1: The histogram of local dimensions estimated near tokens in GPT2 (Robinson et al. [2024]). This histogram shows a mixture with at least three distinct peaks.

Assuming that word embeddings yield manifolds, some researchers have used global dimension estimators on token input embeddings and word embeddings (Kataiwa et al. [2025], Gromov et al. [2024], Tulchinskii et al. [2023]). By using a local (not global) dimension estimator, Robinson et al. [2024] presented the first (to our knowledge) direct test of whether the token subspace is a manifold for the token input embeddings for several LLMs, although no rigorous hypothesis test was presented.

The subspace of tokens has a multimodal distribution of dimension estimates (Figure 1), so unless the space is highly disconnected, it cannot be a manifold.

## 2 Mathematical background

A *manifold* is a space in which every point has a Euclidean neighborhood, and the space can be covered by taking finitely many of these neighborhoods at each point. It follows that if a manifold is connected, then the dimension of each of these Euclidean neighborhoods is the same; we call this the *dimension* of the manifold. Figure 3(a) shows the hollow sphere, which is an example of a 2-dimensional manifold as a subset of $\mathbb{R}^3$.

Given an embedding dimension $\ell$, embedded tokens lie on a *subspace $X$* of Euclidean space $\mathbb{R}^\ell$, with the topology and geometry inherited from $\mathbb{R}^\ell$. Equivalently, one can consider an *embedding function* $e : X \to \mathbb{R}^\ell$, which is one-to-one and has a full rank matrix of partial derivatives (Jacobian matrix). Embedding functions therefore preserve topological structure. The inherited information includes a metric (the Euclidean distance) and a notion of volume. The *volume form* on a subspace is induced by the Euclidean volume. If the subspace is a manifold that is compatible with the inherited metric and volume form, we call the subspace a *Riemannian manifold*. For a 2-dimensional Riemannian manifold, the volume form is the familiar notion of area. In general, if a space is a Riemannian manifold of dimension $d$, then the volume $v$ of a ball of radius $r$ centered at any point is given by an equation of the form Gray [1974],

$$v = Kr^d + (\text{correction terms})r^{d+1}, \tag{1}$$

where $K > 0$ is a constant depending on $d$. If the correction terms are large, the manifold is said to have *high curvature*.

The definition of a manifold is global: if even one point fails to have a Euclidean neighborhood, then the space cannot be a manifold, and such a point is called a *singularity*. One commonly encountered singularity is a *boundary*, in which a point has a neighborhood that is topologically equivalent to a half space. Spaces whose only singularities are boundaries are called *manifolds with boundary*. For instance, the solid ball in $\mathbb{R}^3$ is a manifold with boundary. Its set of singularities is the hollow 2-dimensional sphere.

Classifying the types of singularities up to topological equivalence of their neighborhoods is in general an open problem, though some types are common. Figure 3(b) shows a space with *cusp* and *boundary* singularities. Figure 3(c) shows a singularity associated with a change in dimension.

In order to handle singularities, we also consider a generalization of a manifold called a *fibered manifold with boundary* (*fiber bundle* for short). This is a space in which every point has a neighborhood that can be written as a Cartesian product $B \times F$, where $B$ is a manifold and $F$ is a manifold with boundary. We call $B$ the *base space* and $F$ the *fiber space*.

Curvature (the correction terms in Equation (1)) does not generalize cleanly to spaces with singularities. Instead, the *embedded reach* plays a similar role in what follows. The *embedded reach* (simply *reach*) for a space $X \subseteq \mathbb{R}^\ell$ is the largest distance $\tau \geq 0$ that a point $y \in \mathbb{R}^\ell$ can be from $X$, yet still have a unique closest point on $X$. For convex subspaces, $\tau = \infty$. Conversely, spaces with a cusp (like Figure 3(b)), have $\tau = 0$. Manifolds with small reach have high curvature, and vice versa (Berenfeld et al. [2022]).

## 3 Theoretical insights

Theorem 1 asserts that for a fiber bundle, the number of tokens within a ball of a given radius $r$ centered at a fixed token $\psi$ is a piecewise log-log-linear function of radius, in which the slopes correspond to dimension, and the slopes cannot increase as $r$ increases.

### 3.1 Volume versus radius for fiber bundles

The learned token embedding vectors in an LLM each have the same number of elements, which we refer to as $\ell$. Thus, the explicit representation of the token subspace $T$ used by an LLM arises by an embedding function $e : T \to \mathbb{R}^\ell$ into the *latent space* $\mathbb{R}^\ell$. Since embeddings are one-to-one, each

token $\psi \in T$ corresponds to a unique element $e(\psi) \in \mathbb{R}^\ell$ in the latent space. With a slight abuse of notation, we can write $\psi$ for both.

**Theorem 1.** *Suppose that $T$ is a compact, finite-dimensional Riemannian manifold with boundary, with a volume form $v$ satisfying $v(T) < \infty$, and let $p : T \to S$ be a fiber bundle. If $e : T \to \mathbb{R}^\ell$ is a smooth embedding with reach $\tau$, then there is a function $\rho : e(T) \to [0, \tau]$ such that if $\psi \in e(T)$, the induced volume $(e_* v)$ in $\mathbb{R}^\ell$ satisfies*

$$(e_* v)\left(B_r(\psi)\right) = \begin{cases} O\left(r^{\dim T}\right) & \text{if } 0 \le r \le \rho(\psi), \\ (e_* v)\left(B_{\rho(\psi)}(\psi)\right) + O\left((r - \rho(\psi))^{\dim S}\right) & \text{if } \rho(\psi) \le r, \end{cases} \tag{2}$$

*where $B_r(\psi)$ is the ball of radius $r$ centered at $\psi$, and the asymptotic limits are valid for small $r$.*

*Proof.* See Section A.5 for full details; a sketch follows. Taking the logarithm of both sides of Equation (1) yields a linear equation, in which $d$ is the slope.

For the case of the fiber bundle, Equation (1) must be modified to handle the case where the ball of radius $r$ "sticks out" of the space. This causes a reduction in volume, hence a smaller slope. $\qquad \square$

The probability distribution of tokens in $T$ can be modeled by the volume form $v$. If a fiber bundle is the correct representation of the space of tokens, Theorem 1 characterizes the resulting probability distribution in $\mathbb{R}^\ell$ using parameters (the exponents in Equation (2)) that can be estimated from the token input embedding. These parameters are bounded by the dimensions of $S$ and $T$. In Algorithm 1, Monte Carlo estimates of volume are used, simply by counting tokens within a given radius. Although the convergence of Monte Carlo estimates is a strong assumption, it is also extremely well-established in the literature.

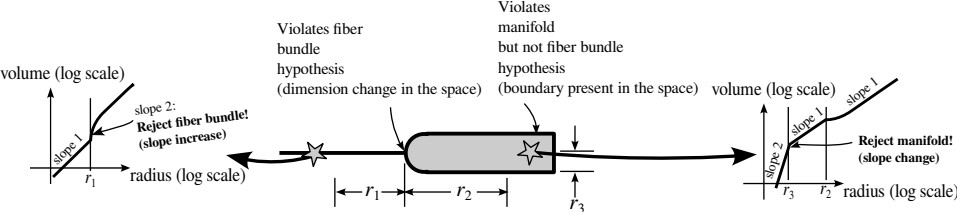

Figure 2: Interpretation of the manifold and fiber bundle tests: the manifold test detects any slope changes in the log volume versus log radius curves, while the fiber bundle test only detects slope increases.

In short, if we take the logarithm of both the volume and radius in Equation (2), we obtain a linear curve for a manifold. Abrupt slope changes imply that the space is not a manifold. For a fiber bundle, we obtain a piecewise linear curve, in which the slopes are given by $\dim S < \dim T$, and the slopes must decrease through a discontinuity as the radius increases, as shown in Figure 2.

### 3.2 Context does not persistently resolve singularities

The remaining question is whether singularities can persist into the *output* of the LLM. Can the user elicit the instabilities caused by singularities? We provide sufficient conditions for unstable outputs:

**Theorem 2.** *Let $Z$ be a $d$-dimensional bounding manifold[1] for the token subspace, such that $T \subseteq Z$. Consider an LLM with a context window of size $w$, in which the latent space of tokens is $\mathbb{R}^\ell$, and we collect $m$ tokens as output from this LLM.*

*Suppose the following, (enough tokens are collected from the response) $m > \frac{2wd}{\ell}$, but (the context window is longer than the number of tokens we collected) $w \ge m$. Under these conditions, a generic set of transformers yields a topological embedding of $T^w = T \times \cdots T$ into the output of the LLM.*

*Proof.* See Section A.5. $\qquad \square$

---

[1]Since $T \subseteq \mathbb{R}^\ell$, it is always contained within such a bounding manifold $Z \subseteq \mathbb{R}^\ell$.

As a consequence, there always exists such an $m$ satisfying the hypothesis if $2d < \ell$, i.e. the bounding manifold has dimension less than half the latent space dimension. If our hypothesis tests show that the token subspace contains singularities and the hypotheses of Theorem 2 are satisfied, singularities will persist into the output of the LLM for generic transformers even for large context window sizes.

## 4  Methods

The tests we propose are implemented via Algorithm 1. The coordinates of the points (token embedding vectors, in Section 5.2) are fixed when we analyze the data, and they do not change from run to run. Algorithm 1 is fully deterministic, so there is no requirement of multiple runs. Centered at a given point, the sorted list of distances to all other points contains the radii at which the Monte Carlo estimate of the volume of a ball increases. The corresponding volumes are simply the indices of this list. Both tests estimate dimension as slope in the log-log progression of the entries versus the indices (3-point centered difference estimates via `numpy.gradient()`), and detect slope changes.

---

**Algorithm 1** Manifold and fiber bundle tests

---

**Require:** $x_1, \ldots, x_n \in \mathbb{R}^\ell$: coordinates for each point
**Require:** $v_{min}$ and $v_{max}$: minimum and maximum number of tokens in neighborhood
**Require:** $W$: sliding window size
**Require:** $\alpha$: significance level
**Ensure:** $p_1$: set of $p$ values for manifold hypothesis
**Ensure:** $p_2$: set of $p$ values for fiber bundle hypothesis
**Ensure:** Set of dimension estimates
 1: **procedure** MANIFOLDANDFIBERBUNDLETEST($x_\bullet, v_{min}, v_{max}, W$)
 2:     Compute $n \times n$ pairwise distance matrix $D$ between all tokens
 3:     **for** Each column of $D$ **do**                    ▷ Columns correspond to token indices
 4:         Sort the column      ▷ Now row indices of distance matrix are volumes, entries are radii
 5:         Retain rows $v_{min}$ through $v_{max}$
 6:         Compute log-log slopes (= dimension estimates) along the column
 7:         Run two sample $T$-test along adjacent sliding windows of size $W$ with level $\alpha$:

   **Manifold test:** Append to $p_1$: the $p$-value for the hypothesis that the slope is constant

   **Fiber bundle test:** Append to $p_2$: the $p$-value for the hypothesis that the slope decreases with row index
 8:         Store both $p$ values and slope with corresponding token (column index)
 9:     **end for**
10:     Apply Holm-Bonferroni multiple test correction to both sets of $p$-values
11: **end procedure**

---

## 5  Results

To demonstrate the validity of our tests in practice, we demonstrate them on three synthetic datasets in Section 5.1 shown in Figure 3(a)–(c). We then turn to the main thrust of the paper, testing LLM token embeddings in Section 5.2. Source code is available at Robinson [2025].

### 5.1  Synthetic datasets

Figure 3(a) shows the result of applying our manifold hypothesis test to the unit sphere in $\mathbb{R}^3$. Since the sphere is a manifold, the $p$ value for every point tested is close to 1, and no rejections of either hypothesis occur.

Figure 3(b) shows the result of applying the manifold test to a portion of the surface given by

$$z^8 + x^2 + y^2 = 0, \text{ where } x^2 + y^2 \leq 1.$$

This surface is a fiber bundle in which the base space is simply the origin, and the fiber spaces are the radial line segments tracing out from the origin to the edge of the disk. It is a manifold with boundary (not a manifold), and the origin $x = y = z = 0$ is a *cusp* singularity. The number of points

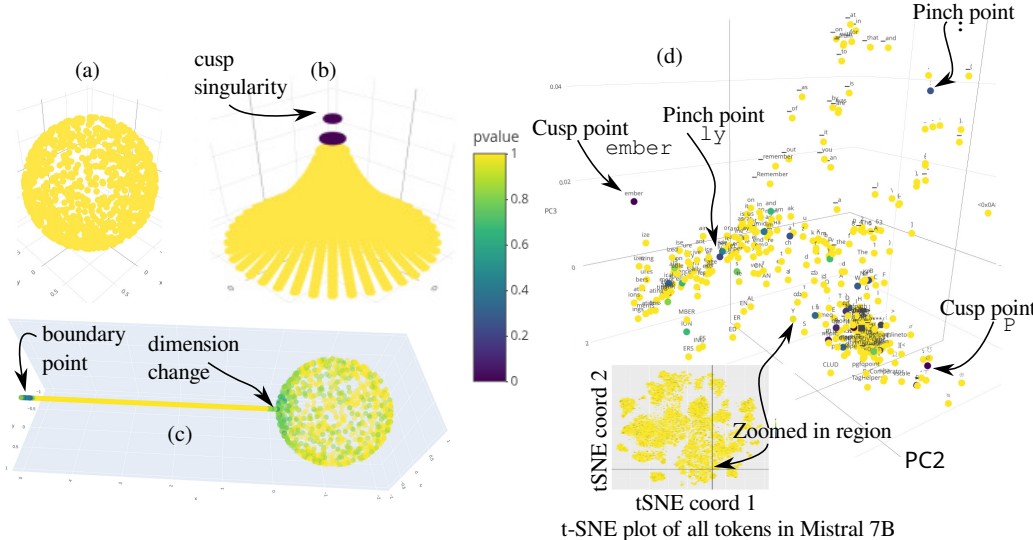

Figure 3: This figure shows the application of this paper's manifold hypothesis test on three synthetic datasets: (a) a manifold, the unit sphere in $\mathbb{R}^3$, (b) a fiber bundle, the cusp surface $z^8 + x^2 + y^2 = 0$ in $\mathbb{R}^3$, (c) neither, a "lollipop" space, and (d) a neighborhood of the token `ember` in Mistral7B. Points are colored by the $p$ values of the manifold hypothesis test; darker colors mean reject the manifold hypothesis.

participating in the cusp and boundary is small (41 points) compared with the total number of points in the dataset (1200 points).

The manifold hypothesis test strongly detects the cusp singularity in Figure 3(b), with $p < 5 \times 10^{-8}$. The manifold test fails to detect the boundary ($p > 0.001$). While using a larger $v_{max}$ in Algorithm 1 improves the test's sensitivity to the point where the boundary is detected, it also reduces the specificity of the test for identifying which points are singularities. On the other hand, the fiber bundle test is not rejected at all, which is consistent with the space being a fiber bundle.

Finally, Figure 3(c) shows a sample of 3000 points drawn from a space shaped like a candy "lollipop," constructed by attaching a line segment (the "stick") to a point on the surface of a hollow sphere (the "candy"). The lollipop is neither a manifold nor a fiber bundle. On the interior of the stick, the space is a 1-dimensional manifold. The candy is a 2-dimensional manifold, except where the stick attaches. The entire space has exactly two singular points. At the very end of the stick is a singular point (a boundary point). Where the stick attaches to the candy is another singularity; it is a transition from the 1-dimensional stick to the 2-dimensional candy.

Running the manifold hypothesis test on the lollipop reveals low $p$-values on points near the boundary of the stick (75 points) and at the transition (58 points) (both have $p < 10^{-5}$). Due to random sampling, the exact points of singularity are not actually present in the dataset; these detections are near the two singularities. None of the points near the boundary point of the stick cause the fiber bundle hypothesis to be rejected with $p < 10^{-5}$, while 2 points near the transition cause the fiber bundle hypothesis to be rejected with $p < 10^{-5}$. Since even one rejection is sufficient, the tests correctly identify that the lollipop is neither a manifold nor a fiber bundle.

## 5.2 LLM token embeddings

We applied our Algorithm 1 to the token subspaces of the four open source LLMs introduced in Sec. 1.1: GPT2, Llemma7B, Mistral7B, and Pythia6.9B. In each LLM we tested, Table 2 exhibits evidence that *the token subspace is not globally a manifold because the local dimension is highly variable ("dim." columns). Moreover, we can infer that they are either not fiber bundles or have high curvature locally*; in either case—violating the hypotheses or having high curvature—model behavior near certain tokens will be highly unstable (low $p$-values in the "rejects" columns). Tables

5–10 in the Supplementary material list every violation we found. Additionally, Figure 3(d) shows the application of the manifold test to the tokens of Mistral7B[2] in the vicinity of the token `ember`. The manifold hypothesis is rejected at various tokens (low $p$-value, shown in darker color).

For each LLM we studied, we chose two pairs of $v_{min}$ and $v_{max}$ parameters in Algorithm 1: one yielding a small radius neighborhood and one yielding a larger radius neighborhood. These parameters were chosen by inspecting a small simple random sample of tokens, with the aim of identifying where changes in dimension were likely to occur. See Section A.6 for details. This configuration results in three tests: a single test for the manifold hypothesis, the fiber bundle test for small radius, and the fiber bundle test for large radius.

Table 2: Dimensional data and number of tokens rejecting the manifold and fiber bundle hypotheses at two slope changes. $n$ is the number of tokens (vocabulary size) in each model. The $p$-values displayed report the minimum $p$-value for each test, accounting for multiple testing. Quartiles (Q2 is the median) are shown for the distribution of the dimensions estimated from the slopes of the log-volume versus log-radius plots aggregated over all tokens. The "rejects" columns list the number of tokens rejecting each hypothesis. Tables 4–10 in the Supplementary material show each of the manifold and fiber bundle hypothesis violations we found.

| Model | Manifold rejects | Fiber bundle | | | |
| | | Smaller Radius | | Larger radius | |
| | | dim. | rejects | dim. | rejects |
|---|---|---|---|---|---|
| GPT2 $n = 50257$ | 66 $p \approx 3 \times 10^{-8}$ | Q1: 20 Q2: 389 Q3: 531 | 12 $p \approx 9 \times 10^{-6}$ | Q1: 8 Q2: 14 Q3: 32 | 7 $p \approx 3 \times 10^{-8}$ |
| Llemma7B $n = 32016$ | 33 $p \approx 5 \times 10^{-9}$ | Q1: 4096 Q2: 4096 Q3: 4096 | 0 N/A | Q1: 8 Q2: 11 Q3: 14 | 1 $p \approx 3 \times 10^{-4}$ |
| Mistral7B $n = 32016$ | 40 $p \approx 3 \times 10^{-7}$ | Q1: 9 Q2: 48 Q3: 220 | 1 $p \approx 8 \times 10^{-4}$ | Q1: 5 Q2: 6 Q3: 9 | 2 $p \approx 8 \times 10^{-5}$ |
| Pythia6.9B $n = 50254$ | 54 $p \approx 2 \times 10^{-7}$ | Q1: 2 Q2: 108 Q3: 235 | 0 N/A | Q1: 2 Q2: 5 Q3: 145 | 0 N/A |

Table 2 shows the results of our hypothesis tests for the four models we analyzed. The models consequently have token input embeddings with vastly different topology, and all of them exhibit highly significant rejections of the manifold hypothesis. GPT2, Llemma7B and Mistral7B also reject the fiber bundle hypothesis. The rejections of the fiber bundle hypothesis are more frequent in the larger radius region than the smaller radius region, which is consistent with the polysemy interpretation of Jakubowski et al. [2020].

The "rejects" columns of Table 2 give the number of tokens that fail our test (in the top rows of each cell). The listed $p$ is the overall $p$-value after Bonferroni correction. The "dim." column shows the quartiles for the estimates of dimensions, at least for the tokens for which a dimension can reliably be ascribed. Note for each of the tokens listed in Tables 5–10, no dimension estimate is possible.

Since there are many rejections of the manifold hypothesis, we summarize some general trends, obtained by manual inspection of Tables 5–10 in the Supplementary material, to which we direct the reader. The GPT2 tokens at irregularities are tokens that can only appear at the beginning of words. The Pythia6.9B tokens at irregularities are nearly all word fragments or short sequences of text that are quite meaningless on their own. The Llemma7B and Mistral7B tokens at irregularities are a combination of the previous two: either they can only appear at the beginning of words or they are word fragments.

While most of the tokens are not shared between the LLMs, Llemma7B and Mistral7B have identical token sets. The fact that Table 2 shows significant differences between these two models indicates that the structure of the irregularities for these two models is quite different. This alone implies that their response to the same prompt is expected to be markedly different, even without considering the

---

[2]More views of Figure 3(d) are shown in Figure 8 in the Supplement.

differences in their respective transformer stages, corroborating the results shown by Vargas et al. [(to appear].

Table 3: Testing the hypotheses of Theorem 2: Will singularities generically persist into the output? See statement of Theorem 2 for variable definitions. "Small" and "Large" refer to the small and large radius estimates of dimensions from Table 2.

| Model | Latent dim $\ell$ | Bounding dim. $d$ | | Context $w$ | Min. output tokens $m$ such that | | Satisfied? $w \geq m$? | |
|---|---|---|---|---|---|---|---|---|
| | | Small | Large | | Small | Large | Small | Large |
| GPT2 | 768 | 389 | 14 | 1024 | 1038 | 38 | No | Yes |
| Llemma7B | 4096 | 4096 | 11 | 4096 | 8193 | 23 | No | Yes |
| Mistral7B | 4096 | 48 | 6 | 4096 | 97 | 13 | Yes | Yes |
| Pythia6.9B | 4096 | 108 | 5 | 4096 | 217 | 11 | Yes | Yes |

Estimates of the bounding manifold dimension used in Theorem 2 for the four LLMs we studied are in the "dimension" columns of Table 2, with the most stringent being the dimension for smaller radius. Table 3 shows the results of testing the hypotheses of Theorem 2 using the medians (Q2). Mistral7B and Pythia6.9B satisfy the hypotheses of Theorem 2 using the small radius dimension estimates, but GPT2 and Llemma7B do not. All four LLMs satisfy the hypotheses of Theorem 2 if we instead use the larger radius dimension estimates. This means that it is likely that the transformers of Mistral7B and Pythia6.9B are unable to resolve the singularities that we show exist in the token subspace, so singularities will persist in the output of the model. If the true bounding manifold for GPT2 or Llemma7B is smaller than the small radius estimate suggests, these too will likely produce singularities in their output.

## 6 Discussion

The large distribution of local dimension (Figure 1 and Table 2) rejects that the token embedding is a globally smooth manifold for each of the LLMs we studied. Beyond non-constant dimension, we also uncovered token neighborhoods without a well-defined intrinsic dimension, a structure not previously reported in these embeddings. These irregular structures suggest that the topology of the embeddings are more complicated than generally assumed. Consequently, all four token subspaces cannot locally be manifolds with large reach, and three cannot be fiber bundles with large reach. Given the aggregation of training data across domains (and even across natural languages), the potential for internal contradictions (contronyms) and confounders (natural language, tweets, source code), it is not surprising to see that our hypotheses are rejected.

The semantic role of tokens likely contributes to LLM sensitivity of prompts that has been observed in the literature (Schulz et al. [2025], Cui et al. [2025], Zhao et al. [2025a], Zilberman [2025], Sclar et al. [2023], Chao et al. [2023]) and may underscore why *elucidating LLM behavior* remains challenging and open. The manifold hypothesis gave both an intuitive and mathematically grounded explanation for the behaviors of learning machines (Liu et al. [2024], Brahma et al. [2015], Narayanan and Mitter [2010]) yet the instability of LLM outputs strongly suggested that these models were different. Our work gives conclusive empirical evidence against the manifold hypothesis for token embeddings and may offer novel ways to interpret LLM behavior as we are not in a regular setting.

Such irregularities are noticeable in a principal components analysis (PCA) projection of the token subspace (Figures 3(d), 7 and 8) but are extremely hard to interpret a priori. Mathematicians are still classifying singularities in $\mathbb{R}^3$ (Zhu et al. [2024]) implying that the classification problem is far from complete (even in $\mathbb{R}^3$). Intuitively, this means that the use of inner products to perform token de-embeddings by orthogonal projections may not be accomplishing the intended objectives. Even small perturbations within the latent space may result in instabilities in these projections, providing potential explanations for the instabilities in LLM outputs. In addition to token instability, the fact that token subspaces are not manifolds and may have small reach means that the geodesic distance between tokens can also be very unstable. While the distance along geodesics can be defined, it may not correlate with any sense of semantic distance between tokens. Furthermore, as Robinson et al. [2024] found, in most of the models, there are tokens with dimension 0 neighborhoods. These tokens

are theoretically *isolated*, implying that the token subspace is disconnected and hence intrinsically infinitely far from other tokens.

Singularities may arise either as artifacts of the training process or from features of the languages being represented. Consistent with the idea that polysemy may yield singularities (Jakubowski et al. [2020]), several of the tokens where the fiber bundle hypothesis is rejected are clear homonyms. For instance, both "affect" and "monitor" can be used as either nouns or verbs, and their meanings are different in these two roles. A token may also correspond to a pair of homonyms after the addition of a prefix or suffix. A token like "aunder" can be prefixed to yield the word "launder", a word with multiple meanings of opposite sense. Specifically, one can "launder" clothing (positive connotation) or "launder" money (negative connotation). Several other tokens where the fiber bundle hypothesis is violated form words with substantially different meanings or grammatical roles upon adding a prefix or suffix. For instance, "wins" can function as a noun, as a verb, or form the adjective "winsome".

The differences in how the manifold and fiber bundle hypotheses are rejected across different LLMs suggest that the training methodology for each model leaves an indelible fingerprint (or gDNA coined by Vargas et al. [(to appear)]). Making general assertions about LLMs without consideration of their training and evolution is likely fraught. Even between Llemma7B and Mistral7B, which have identical token sets, *prompts likely cannot be "ported" from one LLM to another without significant modifications if they contain tokens near irregularities*, likely a consequence of the *invariance of domain theorem*. This may have serious ramifications for interacting LLMs such as those powering multi-agentic frameworks being deployed today (Gu et al. [2025], Jin et al. [2025], Qian et al. [2025]).

Tokens that begin a word or are a word fragment are often located at an irregularity. Additionally, in Llemma7B (but not Mistral7B) and Pythia6.9B, the tokens with unusually low fiber dimension often contain non-printing or whitespace characters. This suggests that these models are quite sensitive to text layout, perhaps to the exclusion of more semantically salient features in the text. Given our findings, future experiments can be run to explore the impact of irregular tokens on the variability of responses produced by different LLMs.

A natural question is if we can find a token embedding that is everywhere regular. One way to approach an answer is to view the number of tokens in a vocabulary as a hyperparameter. There might be a certain vocabulary size that minimizes the number of singularities given the corpus, architecture, etc. Even the best vocabulary might lead to a space that could still be singular, for reasons inspired by Jakubowski et al. [2020]. Consider the ∗ symbol, which happens to be singular for Mistral7B (See Table 8, number 29). In natural language, it has a very distinctive meaning in terms of calling out footnotes, while in other contexts it can be used for multiplication or marking text as bold (in Markdown). Therefore, this character connects these different semantics. Suppose that each language were in fact well-described by a manifold. Then these manifolds would be joined together in the entire latent space at the ∗ (and likely other points). As a result, this token with high likelihood would be a singularity because the dimensions for different languages are almost surely different (Tulchinskii et al. [2023]). In reality, tokens are neither in 1:1 correspondence with words nor with grammatical symbols. Not only do parts of words map onto a single token, but tokenization is a many to one map across the board. How well context / attention / transformers can unfurl these mappings is a big part of the mystery of how these models work. Identifying and analyzing the irregularities may help us understand the approximate limits of attention/transformers architectures.

As we attempt to model natural phenomena particularly in our physical world, we know singularities exist (e.g. phase transitions), and so assuming regularity limits our understanding of the natural world. Our work extends this notion to the linguistics of natural language where its richness and complexity do not appear to manifest in a smooth representation.

# 7   Limitations

While topological theory motivates our statistical test, only the slope changes that occur for radii less than the embedded reach correspond to rejections of the manifold or fiber bundle hypotheses. If the reach of a manifold is small, then it likely has a high local curvature, and measuring distances on that surface will be numerically unstable. And lastly, we emphasize that these tests allow us to reject the structure assumed in our null hypothesis (smooth and regular), but do not permit us to infer an alternative. While rejecting regularity does imply singularity, it does not afford us any evidence as to the type of singularity. That must be determined *post hoc* if possible.

## Acknowledgments and Funding Disclosure

The authors would like to thank Mohammed Abouzaid, Anand Sarwate, Andrew Lauziere, and Andrew Engel for helpful suggestions on a draft of this manuscript.

This material is based upon work partially supported by the Defense Advanced Research Projects Agency (DARPA) under Contract No. HR001124C0319. Any opinions, findings and conclusions or recommendations expressed in this material are those of the author(s) and do not necessarily reflect the views of the Defense Advanced Research Projects Agency (DARPA). Distribution Statement "A" (Approved for Public Release, Distribution Unlimited) applies to the portion of the work funded by DARPA.

The authors have no competing interests to declare.

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

# A Supplementary Material

## A.1 Additional discussion of the literature

Although there a number of mathematically distinct definitions of dimension, they coincide for manifolds. Intrinsic dimension of a point cloud has been widely studied in the literature Schweinhart [2020]. All of the papers that address practical estimation of intrinsic dimension from data do so under the assumption that the data lie on a manifold. For instance, Levina and Bickel [2004] provides maximum likelihood estimates, operating under the assumption that data "can be efficiently summarized in a space of a much lower dimension, such as a nonlinear manifold." Moreover, Facco et al. [2017] provides a surprisingly effective approach using extremely small neighborhoods, a fact which underlies our use of our test on a "small radius neighborhood" in Section 5.2.

Even though we ultimately reject the manifold hypothesis, the token subspace has been studied under the assumption that it is a manifold. For instance Lee et al. [2025] observe that,"Language models often construct similar local representations as constructed from local linear embeddings." We note that most of the tokens in a typical vocabulary fail to reject the manifold hypothesis, which means that without the deeper analysis we pursue, it is easy to overlook important differences. Furthermore, our Table 2 shows significant topological differences in the local representations between models.

As introduced in Section 1.2, we mentioned that Lee et al. [2025] show a few things that are consistent with our findings. They note that "Token embeddings exhibit low intrinsic dimensions," which agrees with our in Table 2, at least in the large radius neighborhood. Their estimates of dimension for GPT2 is within the confidence interval we computed. They do not recognize that there might be a fiber bundle structure in play, so Lee et al. [2025] does not estimate dimensions from a smaller radius neighborhood.

Relevant to Theorem 2, there is a possibility that the implicit space of possible activations can be inferred from the behavior of the model Zhao et al. [2025b]. Since Theorem 2 represents the context window as a product of copies of the latent space, a projection of this implicit structure is the token subspace. In this vein, Zhao et al. [2025b] is not inconsistent with other findings within the literature, for instance Robinson et al. [2024]. This latter paper shows that the token subspace can be inferred (up to topological equivalence) from the responses of the LLM to single-token prompts. Taken together, Zhao et al. [2025b], Robinson et al. [2024] and the present one have pretty significant implications. Even though the token subspace may have about 0.1% of tokens that are singular, we suspect that a user will be encountering singular tokens on a regular basis when using LLMs (which seems empirically plausible). Estimating the rate of occurrence of a singular token remains an open question.

## A.2 LLM Background

At an abstract but precise level, an LLM consists of several interacting processes, as outlined in Figure 4. An LLM implements a transformation of a sequence of tokens (the query) into a new sequence of tokens (the response). Formally, if each input token is an element of a metric space $T$, then the LLM is a transformation $T^w \to T^m$, where $w$ is the number of tokens in the query and $m$ is the number of tokens in the response. This transformation is typically *not* a function because it is stochastic—it involves random draws.

To operate upon tokens using numerical models, such as could be implemented using neural networks, we must transform the finite set of tokens $T$ into numerical data. This is typically done by way of a pair of *latent spaces* $X = \mathbb{R}^\ell$ and $Y = \mathbb{R}^q$. The dimension $q$ of $Y$ is chosen to be equal to the number of elements in $T$, so that elements of $Y$ have the interpretation of being (unnormalized) probability distributions over $T$.

The transformation $T^w \to T^m$ is constructed in several stages.

**Input tokenization** : Each token is embedded individually via the *token input embedding* function $e : T \to X$. As a whole, $X^w$ is called a *latent window*.

**Transformer blocks** : The probability distribution for the next token is constructed by a continuous function $f : X^w \to Y$. This is usually implemented by one or more *transformer blocks*.

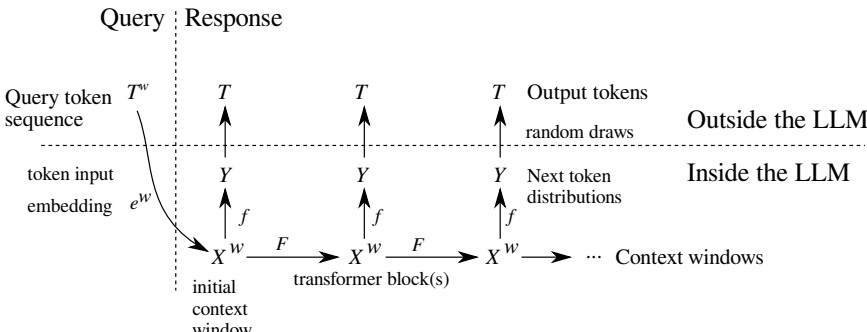

Figure 4: Data flow in a typical LLM. A sequence of tokens forming the query is converted via the token input embedding $e^w$ into the initial context window, as a point in the latent space $X^w$. Each of these windows in the latent space are converted, token-by-token, into probability distributions via $f$ into the single token latent space $X$. From these, each token presented in the output (in the set $Y$) is obtained via a random draw. These output tokens are then used for subsequent windows.

**Output tokenization** : Given the output of one of the transformer blocks $f$, one can obtain an output token in $T$ by a random draw. Specifically, if $(\psi_1, \psi_2, \ldots, \psi_w)$ is the current window in $X^w$, then the next token $t$ is drawn from the distribution given by $f(\psi_1, \psi_2, \ldots, \psi_w)$.

**Next window prediction** : Given that token $t$ was drawn from the distribution, the next latent window itself is constructed by a transformation $F : X^w \to X^w$, which advances the window as follows:
$$F(\psi_1, \psi_2, \ldots, \psi_w) := (\psi_2, \ldots, \psi_w, t).$$

Note well: the function $e^w : T^w \to X^w$ which transforms a sequence of individual tokens into coordinates in the latent space is merely the first stage in the process. As such, $e^w$ *is distinct from* the embedding of sequences provided by many LLM software interfaces. That sequence embedding is properly an intermediate stage within the function $f$.

The focus of this paper is specifically upon the structure of the *token input embedding* $e : T \to X = \mathbb{R}^\ell$. Since the token set $T$ is finite, $e$ can be stored as a matrix. In this matrix, each column corresponds to an element of $T$, and thereby ascribes a vector of numerical coordinates to each token. By replacing the last layer of the deep neural network $f : X^w \to Y$, a vector of probabilities for the next token is obtained from the activations of the last layer. One can therefore interpret the probabilities as specifying a *token output embedding*. Both the tokenization and the transformer stages are learned during training, and many strategies for this learning process are discussed extensively in the literature. These two stages interact when they produce the LLM output, so it is important to understand the lineage of a given tokenization as being from a particular LLM.

### A.3   Computational resource requirements

Applying our entire method (starting with the token embedding matrix and ending with the $p$-values for each of the three tests at every token) to each model took approximately 12 hours of wall clock time (per model) on an **Intel Core i7-3820 with 32 GB of CPU RAM and no GPU running at 3.60GHz**. Almost all of the runtime was spent computing the token pairwise distance matrix $D$ in Algorithm 1. The distance matrix calculation was performed using `scipy.spatial.distance_matrix()`, which produces a dense matrix, by computing the distance between all pairs of tokens in the latent space. Because of the relatively low runtime requirements, we did not feel the need to use sparsity-enhancing techniques to accelerate the distance matrix calculation. The subsequent stages, including the three hypothesis tests run in only a few minutes on the same machine.

### A.4   Interpretation of fiber bundles in terms of spatial variability

It is usual to describe measurements as exhibiting the combined effect of multiple spatial scales. One way this might arise is if we could express each measurement as being an ordered pair (signal, noise), with the short spatial scales correlating to noise and the longer spatial scales correlating to the signal.

With this interpretation, if the space of all possible signals is $S$ and the space of all noise values is $V$, we could represent the space of all possible measurements as the cartesian product $E = S \times V$. This cartesian product representation can arise for noiseless data as well, as it merely captures multiple spatial scales. In what follows, we will call $S$ the *base space* and $V$ the *fiber space*.

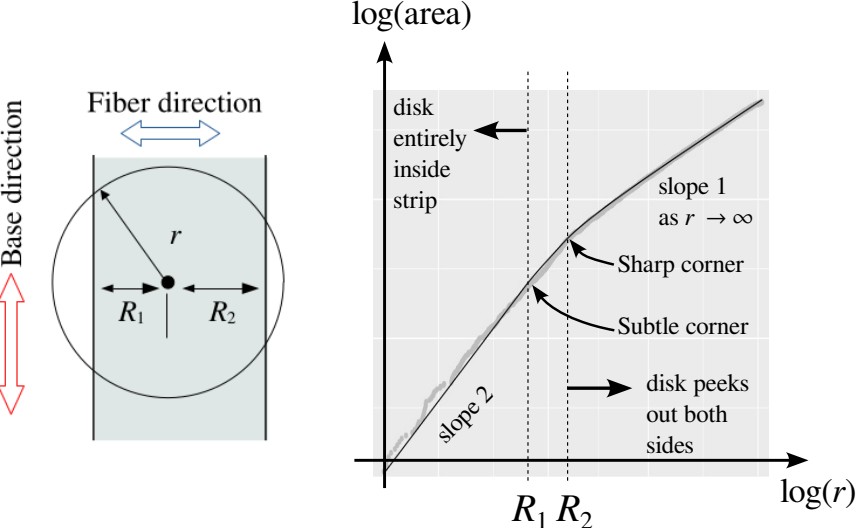

Figure 5: Our method applied to a fiber bundle in $\mathbb{R}^2$. The vertical direction is the base space (long spatial scale), while the horizontal direction represents the fibers space (short spatial scale). Gray points on the right frame show estimates from a random sampling of points in the strip; the solid line shows the theoretical area versus radius curve.

Figure 5 shows an example of this situation. It consists of a 1-dimensional base space (the large radius neighborhood) and 1-dimensional fibers (the small radius neighborhood), which in this case forms a narrow strip in the plane. Volumes (areas, in this case) of balls of small radius scale quadratically (slope 2 in a log-log plot), but scale asymptotically linearly (slope 1 in a log-log plot) for large radii. The transition between these two behaviors is detectable by way of a corner in the plot. This situation is easily and robustly estimated from the data; the gray points in Figure 5 (right) are derived from a random sampling of points drawn from the strip.

To test for multiple spatial scales, we propose a signal model that is mathematically represented by a *fiber bundle*. In a fiber bundle, the long spatial scales are modeled by a space $S$, but all possible measurements are modeled by a function $p : E \to S$. The idea is that *fibers* $p^{-1}(b)$ are still cartesian products: pairs of both spatial scales, and these are all identical up to smooth changes of coordinates. Our method relies upon a particular geometric property of fiber bundles: we can identify if the fibers are not all identical according to when the conclusion of Theorem 1 is violated.

One possible way that a fiber bundle might arise is if the underlying noise model is *homoscedastic*, which is to say that the dimension of the noise near a token does not depend on that token. In this case, the fibers (small radii) mostly correspond to noise, and the base (large radii) correspond mostly to signal. Nevertheless, a fiber bundle may arise with noiseless data as well; the presence of multiple spatial scales is what matters.

On the other hand, if the underlying noise model is *heteroscedastic*—it depends on the token in question—then the fiber bundle model should be rejected. Figure 2 shows a situation that is not a fiber bundle, since there is a change in the dimension of the fiber. In the left portion of the figure, the fiber dimension is $0$ while in the right portion the fiber dimension is $1$. This is detectable by looking at the volume versus radius plots for two samples. While both samples show corners in their volume versus radius plots, Theorem 1 establishes that *the slopes always decrease with increasing radius* for a fiber bundle. This is violated for the sample marked $\psi_1$, so we conclude that the space is not a fiber bundle. On the other hand, because the sample marked $\psi_2$ does not exhibit this violation, it is important to note that if a sample yields data consistent with Theorem 1, *we cannot* conclude that the space is a fiber bundle.

## A.5 Mathematical proof of Theorems 1 and 2

This section contains mathematical justification for the fiber bundle hypothesis proposed earlier in the paper and the proof of Theorems 1 and 2. The central idea is the use of a special kind of fiber bundle, namely a fibered manifold with boundary.

By the submersion theorem (Lee [2003]), if the Jacobian matrix of a fibered manifold $p : T \to S$ at every point has rank equal to the dimension of $S$, then the preimages $p^{-1}(\psi) \subseteq T$ of each point $\psi \in T$ are all diffeomorphic to each other. These preimages form the *fibers* discussed in the earlier sections of the paper.

As a consequence, each point $y$ in the base space $S$ has an open neighborhood $U$ where the preimage $p^{-1}(U)$ is diffeomorphic to the product $U \times p^{-1}(y)$, which is precisely the base-fiber split discussed in Section A.4. Specifically, the base dimension is simply the dimension of $S$, whereas the fiber dimension is the $(\dim T - \dim S)$.

The notion of a fibered manifold $p : T \to S$ forms the intrinsic model of the data, which is only implicit in an LLM. The tokens present in a given LLM can be thought of as a sample from a probability distribution $v$ on $T$, which can be taken to be the Riemannian volume form on $T$ normalized so that $v(T) = 1$.

**Definition 1.** If $e : T \to \mathbb{R}^{\ell}$ is a continuous map and $v$ is a volume form on $T$, then the *pushforward* is defined by

$$(e_* v)(V) := v \left( e^{-1}(V) \right)$$

for each measurable set $V$.

It is a standard fact that if $e$ is a fibered manifold or an embedding, then $e_* v$ is also a volume form.

*Proof.* (of Theorem 1) If $e$ is assumed to be a smooth embedding[3], the image of $e$ is a manifold of dimension $\dim T$. The pushforward of a volume form is a contravariant functor, so this means that $e_* v$ is the volume form for a Riemannian metric on $e(T)$. Using this Riemannian metric on $e(T)$, then [Gray, 1974, Thm 3.1] implies that for every $\psi \in e(T)$, if $r \ll \tau$, then

$$(e_* v) \left( B_r(\psi) \right) = O \left( r^{\dim T} \right). \tag{3}$$

Since $T$ is compact, $S$ is also compact via the surjectivity of $p$. This implies that there is a maximum radius $r_1$ for which a ball of this radius centered on a point on $\psi \in e(T)$ is entirely contained within $e(T)$. Also by compactness of $S$, there is a minimum radius $r_2$ such that a ball of radius $r_2$ centered on a point $\psi \in e(T)$ contains a point outside of $e(T)$.

Since $e$ is assumed to be an embedding, by the tubular neighborhood theorem (Lee [2003]), it must be that $r_2 < \tau$. Define

$$\rho(\psi) := \operatorname{argmax}_r \left\{ B_r(\psi) \subseteq e(T) \right\},$$

from which it follows that $0 < r_1 \leq \rho(\psi) \leq r_2 < \tau$. As a result, Equation (3) holds for all $r \leq \rho(\psi)$, which is also the first case listed in Equation (2).

If $r$ is chosen such that $\rho(\psi) < r < \tau$, the volume of the ball centered on $\psi$ of radius $r$ will be less than what is given by Equation (3), namely

$$(e_* v) \left( B_r(\psi) \right) < O(r^{\dim T}).$$

Since $v$ is a volume form, its pushforward $(p_* v)$ onto $S$ is also a volume form. Moreover, via the surjectivity of $p$,

$$\begin{aligned}
(e_* v)(B_r(\psi)) &= v(e^{-1}(B_r(\psi))) \\
&\leq v(p^{-1}(p(e^{-1}(B_r(\psi))))) \\
&\leq (p_* v)(p(e^{-1}(B_r(\psi)))) \\
&\leq O(r^{\dim S}).
\end{aligned}$$

From this, the second case of Equation (2) follows by recentering the asymptotic series on $\rho(\psi)$. $\square$

---

[3]Smoothness is not a strong constraint here. Any continuous function (such as ReLU) can be approximated well enough by a smooth map.

Notice that the second case in Equation (2) may be precluded since while it holds for small $r$, it may be that $\rho(\psi)$ may not be sufficiently small. As a consequence, the second case only occurs when both $r$ and $\rho(\psi)$ are sufficiently small. In the results shown in Section 5.2, both cases appear to hold frequently.

We now address Theorem 2. Informally, we cannot get rid of all stochasticity during training (due to finite precision, stochastic gradient descent, etc.). This ensures that the transformers in practice are always in general position, hence *generic*.

Given a context window of $w - 1$ tokens, an LLM uses its learned parameters to predict the next ($w$-th) token. This gets added to the context window and the process continues. (See Section A.2 for details.) In this section, we provide a mathematical argument showing that the use of context is insufficient to resolve singularities present in the token subspace for a generic transformer in a persistent manner. Again, denote the token subspace by $T$, noting that Section 5.2 shows that $T$ is unlikely to be a manifold for typical LLMs. Let us begin with a lemma:

**Lemma 1.** *If $X$ is a manifold, $Y$ is an arbitrary topological space, and $X \times Y$ is a manifold, then $Y$ must also be a manifold.*

*Proof.* Observe that $Y$ is homeomorphic to $(X \times Y)/X$, which is the quotient of two manifolds via the projection map, which is a surjective submersion. Hence $Y$ is a manifold. □

Consequently, the contrapositive of Lemma 1 states that if $T$ is not a manifold but $T^{w-1}$ is a manifold, it follows that $T^w$, the space of sequences of tokens, cannot be a manifold. At best, the latent space for every other context window size can be a manifold. For every proposed context window length, a longer window will exhibit singularities. In short, *lengthening the context window does not persistently resolve singularities* in the token embedding.

For instance, if we use the embedding interface provided by the LLM to embed a sequence of $w$ tokens, we will obtain a vector in the latent space $\mathbb{R}^\ell$. On the other hand, if we embed each of the tokens individually yet sequentially, we obtain a vector in $\mathbb{R}^{\ell \times w}$. The transformer uses the vector in $\mathbb{R}^{\ell \times w}$ to determine the vector in $\mathbb{R}^\ell$. Theorem 2 states that the latter ($\mathbb{R}^{\ell \times w}$) is crucial in understanding the output of the LLM. (See Section A.2 for further details.)

*Proof.* (of Theorem 2) We need to manipulate the inequalities in the hypothesis to show that they satisfy [Robinson et al., 2025, Thm. 1]. Once the hypothesis is satisfied, [Robinson et al., 2025, Thm. 1] establishes that a generic set of transformers will embed $T^w$ into the output of the LLM. Notice that [Robinson et al., 2025, Thm. 1] considers the response of the LLM to a single token, though this can be easily extended by enlarging the bounding manifold dimension. With this in mind, the hypothesis to be satisfied is that

$$2wd < m\ell \leq w\ell.$$

The right inequality is clearly satisfied if context window is longer than number of output tokens collected, namely $w \geq m$. The left inequality is satisfied if

$$2d < \frac{m}{w}\ell,$$

which can be easily rearranged to the form in the hypothesis. □

### A.6 Case study of some tokens in GPT2

Figure 6 demonstrates our method on three tokens used by GPT2. The tokens \$ and # play an important role in many programming languages, whereas ¢ does not. (Note that these tokens were chosen for illustrative purposes. The $p$-value for rejecting the hypotheses is larger than $\alpha = 10^{-3}$ used in Section 5.2, so these particular tokens do not appear in any of the tables in the following sections.) The token # does not show any rejections of our hypotheses. While this does not allow one to conclude that the vicinity of # is a manifold or fiber bundle, we can use Theorem 1 to estimate the dimension from the slope of the curve (approximately 53, once logarithms of the values on both axes are taken). The token ¢ exhibits a rejection of the manifold hypothesis, but not the fiber bundle hypothesis. We can conclude that there is a irregularity at a distance of roughly 3.9 units of ¢. The curve for \$ exhibits two slope changes at radii 3.8 and 4.0. The smaller radius slope change represents

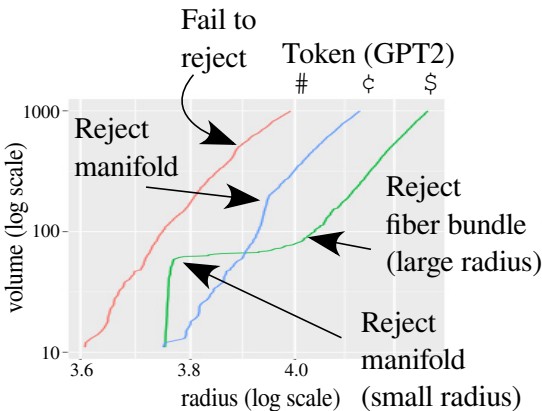

Figure 6: Application of the tests to real data consisting of the tokens #, ¢, and $ with GPT2's token input embedding. Notice that in the case of $, the first slope change decreases the slope; the fiber bundle test fails if the slope increases.

a violation of the manifold hypothesis, while the large radius slope change represents a violation of the fiber bundle hypothesis.

Given the presence of multiple slope changes, it is necessary to select the parameters $v_{min}$ and $v_{max}$ in Algorithm 1 carefully. Ideally, one aims to capture at most one slope change between $v_{min}$ and $v_{max}$. This is not specifically required by Algorithm 1, since it uses a sliding window, multiple rejections may be detected.

## A.7  Case study of a neighborhood of `ember` in Mistral7B

Figures 7 and 8 show the application of this paper's manifold hypothesis test to a neighborhood of the token `ember` in Mistral7B. Since the latent space of Mistral7B is $4096$, we have projected the coordinates of each token shown to $3$ using PCA. The inset shows all of the tokens in Mistral7B (rendered with t-SNE), with a crosshairs showing the location of the zoomed-in region. Each point shown in Figure 7 is a token used by Mistral7B, colored based upon the $p$-value for the manifold hypothesis test. Darker colors correspond to lower $p$-values, which are less consistent with the hypothesis that the token's neighborhood is a manifold with low curvature.

Given that the we have established that manifold rejections occur in Section 5.2, we can attempt to understand possible causes. Intuitively, several non-manifold features are visibly present[4] in Figure 7. The tokens "`ember`" and "`P`" appear to be both putative examples of *cusp points*, places where the space exhibits a sharp exposed point. Manifolds cannot have cusp points, so attempting to approximate a cusp with a manifold—for instance, with UMAP—necessitates high curvature. The tokens "`:`" and "`ly`" appear to be *pinch points*—like what are present in a chain of sausages—which again cannot be approximated by a manifold unless it has high curvature.

## A.8  Table of fiber bundle rejections in all four LLMs

See Table 4.

## A.9  Tables of manifold rejections in all four LLMs

See Tables 5–10.

---

[4]The coordinates, token text, $p$-values, and a video of the space with commentary are available in the ZIP file Supplement. Due to size requirements for the coordinates of the entire vocabularies for all four LLMs, we will upload the coordinates to a public repository.

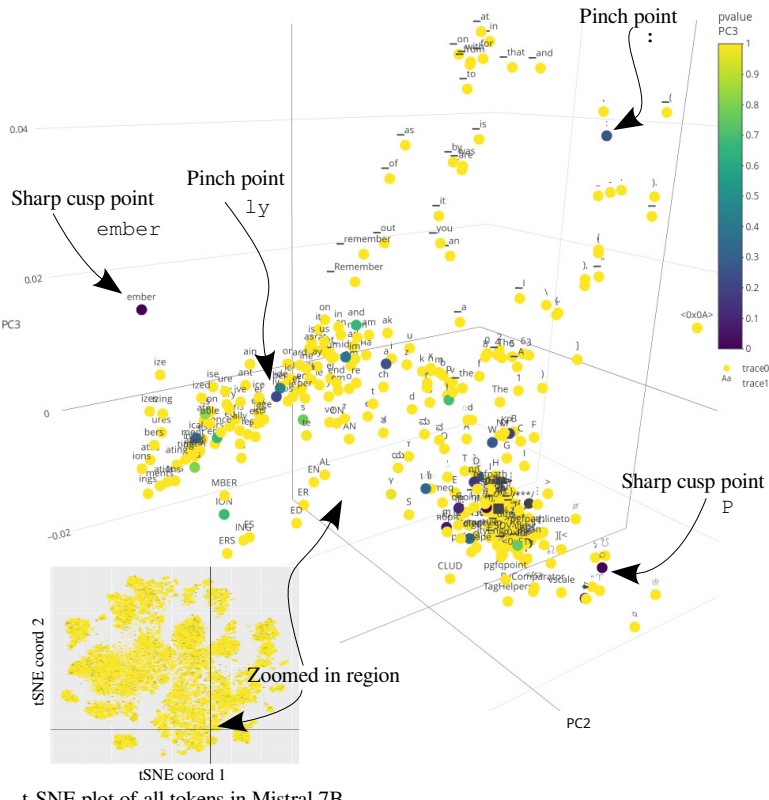

Figure 7: This figure shows the application of this paper's manifold hypothesis test to a neighborhood of the token `ember` in Mistral7B. (The fiber bundle tests are not shown due to space considerations.) Since the latent space of Mistral7B has dimension $\ell = 4096$, we have projected the coordinates of each token shown to 3 using PCA. The inset shows all of the tokens in Mistral7B (rendered with t-SNE), with a crosshairs showing the location of the zoomed-in region. Each point shown in this figure is a token used by Mistral7B, colored based upon the $p$-value for the manifold hypothesis test. Darker colors correspond to lower $p$-values, which are less consistent with the hypothesis that the token's neighborhood is a manifold with low curvature. Putative geometric structures are labeled for certain tokens.

## A.10   Limitations of theory

Because our approach requires the distances between all pairs of tokens, to use the hypothesis tests presented in this paper, one needs to have access to the token embedding vectors. This is why our paper considers four open source models of moderate size – GPT2 (Radford et al. [2019]), Llemma7B (Azerbayev et al. [2024]), Mistral7B (Jiang et al. [2023]), and Pythia6.9B (Biderman et al. [2023]). While Robinson et al. [2025] does provide an approach to extract the token embeddings using systematic prompting, which would allow the application of our tests to proprietary models, their method requires exquisite control of the prompting and is computationally expensive.

The reach itself is difficult to estimate accurately from sampled data (Berenfeld et al. [2022], Rawson and Robinson [2021], Aamari and Levrard [2019]). Our implementation tests the radius below an estimated reach, determined by an inspection of the radius-vs-volume plots of a random sample of tokens. If the true reach is smaller than our estimates, the curvature of the manifolds that fit the data well, found by *any* method, will be high.

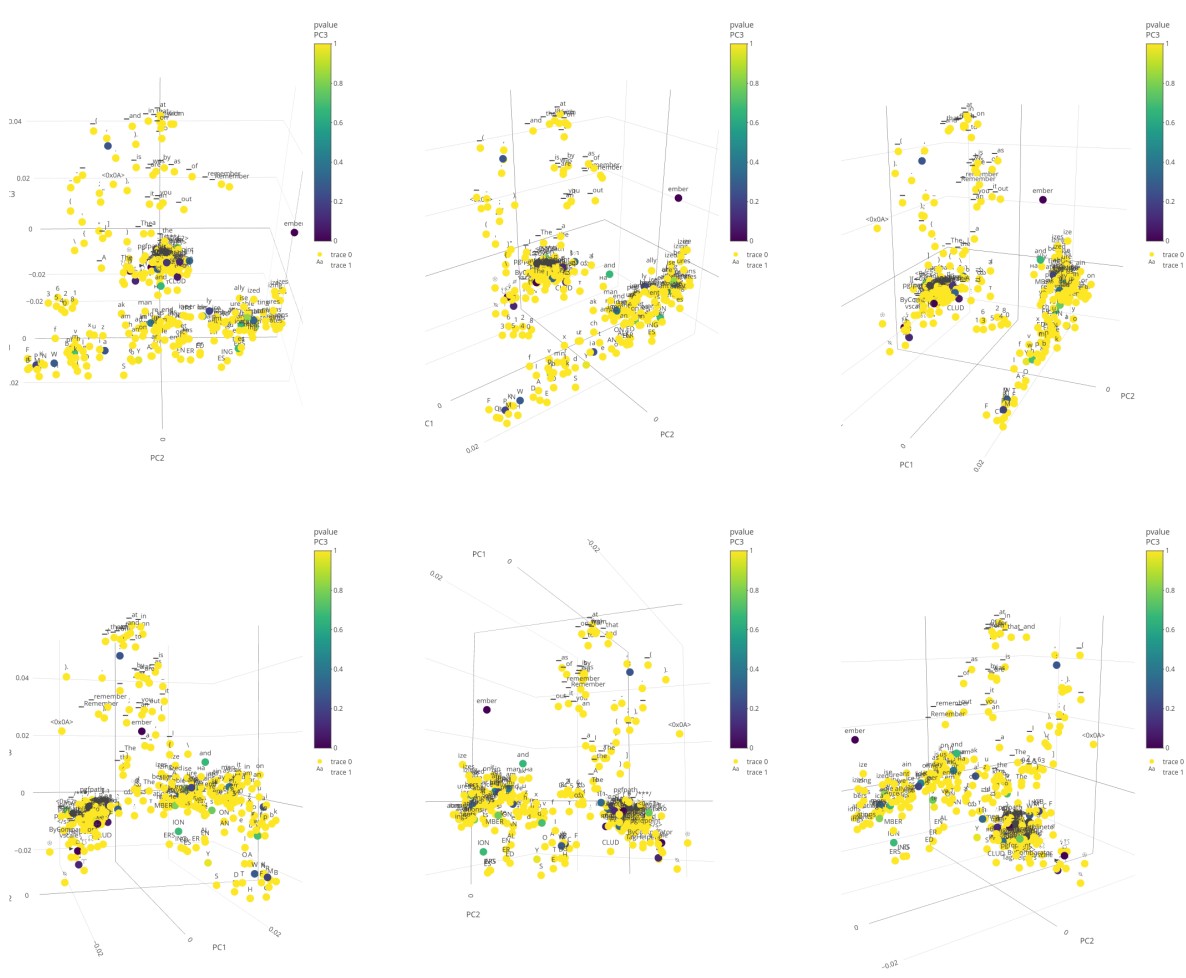

Figure 8: Additional rotational views of the PCA of the neighborhood of `ember` in Mistral7B shown in Figure 7. Each point shown is a token. Colors are $p$-values for the manifold hypothesis, with darker colors corresponding to lower likelihood that that point's neighborhood is a manifold with low curvature.

Table 4: Violations of the fiber bundle hypothesis

| Model | Token ID | Token | Slope change | $p$-value | Comment |
|---|---|---|---|---|---|
| GPT2 | 47623 | `laughable` | Smaller radius | $9 \times 10^{-6}$ | Must start a word |
| GPT2 | 47128 | `nuance` | Smaller radius | $2 \times 10^{-4}$ | Must start a word |
| GPT2 | 28664 | `dt` | Smaller radius | $2 \times 10^{-4}$ | |
| GPT2 | 37031 | `Mesh` | Smaller radius | $2 \times 10^{-4}$ | |
| GPT2 | 2689 | `affect` | Smaller radius | $3 \times 10^{-4}$ | Must start a word |
| GPT2 | 48387 | `Thankfully` | Smaller radius | $3 \times 10^{-4}$ | |
| GPT2 | 44284 | `swat` | Smaller radius | $6 \times 10^{-4}$ | Must start a word |
| GPT2 | 30031 | `Malaysian` | Smaller radius | $6 \times 10^{-4}$ | Must start a word |
| GPT2 | 8501 | `Palestinian` | Smaller radius | $7 \times 10^{-4}$ | Must start a word |
| GPT2 | 7864 | `wins` | Smaller radius | $8 \times 10^{-4}$ | Must start a word |
| GPT2 | 46086 | `hedon` | Smaller radius | $9 \times 10^{-4}$ | |
| GPT2 | 17052 | `donor` | Smaller radius | $9 \times 10^{-4}$ | Must start a word |
| GPT2 | 47482 | `Xan` | Larger radius | $3 \times 10^{-8}$ | Must start a word |
| GPT2 | 21118 | `aunder` | Larger radius | $2 \times 10^{-4}$ | |
| GPT2 | 20564 | `Dri` | Larger radius | $2 \times 10^{-4}$ | |
| GPT2 | 1681 | `ney` | Larger radius | $3 \times 10^{-4}$ | |
| GPT2 | 2076 | `rodu` | Larger radius | $3 \times 10^{-4}$ | |
| GPT2 | 44402 | `Insert` | Larger radius | $4 \times 10^{-4}$ | |
| GPT2 | 38833 | `Ying` | Larger radius | $4 \times 10^{-4}$ | Must start a word |
| Llemma7B | 16352 | `pax` | Larger radius | $3 \times 10^{-4}$ | |
| Mistral7B | 25219 | `änge` | Smaller radius | $8 \times 10^{-4}$ | |
| Mistral7B | 9009 | `monitor` | Larger radius | $8 \times 10^{-5}$ | Must start a word |
| Mistral7B | 4104 | `H0` | Larger radius | $5 \times 10^{-4}$ | |

Table 5: Rejections of the manifold hypothesis for GPT2 (1 of 2)

| Number | Token ID | Token | $p$-value | Comment |
|---|---|---|---|---|
| 1 | 47482 | Xan | 0.0000000280 | Must start a word |
| 2 | 28899 | chemist | 0.0000000602 | |
| 3 | 32959 | fry | 0.00000564 | Must start a word |
| 4 | 47623 | laughable | 0.00000807 | Must start a word |
| 5 | 29884 | Retro | 0.0000108 | Must start a word |
| 6 | 48387 | Thankfully | 0.0000339 | |
| 7 | 20619 | intoler | 0.0000514 | Must start a word |
| 8 | 18533 | assumes | 0.0000533 | Must start a word |
| 9 | 45916 | handwritten | 0.0000783 | Must start a word |
| 10 | 17615 | MY | 0.0000811 | Must start a word |
| 11 | 47128 | nuance | 0.000105 | Must start a word |
| 12 | 24817 | playground | 0.000110 | Must start a word |
| 13 | 28664 | dt | 0.000124 | |
| 14 | 21118 | aunder | 0.000132 | |
| 15 | 37031 | Mesh | 0.000135 | |
| 16 | 37338 | Disney | 0.000139 | |
| 17 | 23780 | McCl | 0.000155 | Must start a word |
| 18 | 20564 | Dri | 0.000196 | |
| 19 | 17886 | Destiny | 0.000206 | Must start a word |
| 20 | 1681 | ney | 0.000206 | |
| 21 | 21904 | juvenile | 0.000209 | Must start a word |
| 22 | 19980 | advent | 0.000211 | Must start a word |
| 23 | 8939 | perpet | 0.000213 | Must start a word |
| 24 | 38908 | otally | 0.000223 | |
| 25 | 2689 | affect | 0.000226 | Must start a word |
| 26 | 16814 | efficient | 0.000238 | |
| 27 | 2076 | rodu | 0.000242 | |
| 28 | 14770 | Kl | 0.000260 | Must start a word |
| 29 | 2359 | duct | 0.000284 | |
| 30 | 41890 | frail | 0.000292 | Must start a word |
| 31 | 28654 | alle | 0.000304 | Must start a word |
| 32 | 20703 | Cha | 0.000322 | Must start a word |
| 33 | 6684 | OO | 0.000333 | |

Table 6: Rejections of the manifold hypothesis for GPT2 (2 of 2)

| Number | Token ID | Token | $p$-value | Comment |
|---|---|---|---|---|
| 34 | 44402 | Insert | 0.000336 | |
| 35 | 38833 | Ying | 0.000357 | Must start a word |
| 36 | 41072 | Georgia | 0.000373 | |
| 37 | 48679 | flo | 0.000384 | |
| 38 | 32745 | uddled | 0.000413 | |
| 39 | 6399 | subsequ | 0.000421 | Must start a word |
| 40 | 26106 | lig | 0.000439 | Must start a word |
| 41 | 13272 | Environmental | 0.000456 | Must start a word |
| 42 | 14840 | Vladimir | 0.000466 | Must start a word |
| 43 | 16721 | Defence | 0.000510 | Must start a word |
| 44 | 44284 | swat | 0.000558 | Must start a word |
| 45 | 28336 | registering | 0.000564 | Must start a word |
| 46 | 30031 | Malaysian | 0.000569 | Must start a word |
| 47 | 45047 | trough | 0.000575 | Must start a word |
| 48 | 11368 | ears | 0.000615 | Must start a word |
| 49 | 37070 | Meh | 0.000637 | Must start a word |
| 50 | 8501 | Palestinian | 0.000648 | Must start a word |
| 51 | 22053 | reckless | 0.000654 | Must start a word |
| 52 | 42656 | Wax | 0.000658 | Must start a word |
| 53 | 4307 | Dist | 0.000673 | Must start a word |
| 54 | 20648 | Quad | 0.000725 | Must start a word |
| 55 | 39468 | relations | 0.000749 | |
| 56 | 7864 | wins | 0.000781 | Must start a word |
| 57 | 42839 | KER | 0.000785 | |
| 58 | 50057 | gist | 0.000786 | Must start a word |
| 59 | 46086 | hedon | 0.000834 | |
| 60 | 19055 | MMA | 0.000839 | Must start a word |
| 61 | 17052 | donor | 0.000865 | Must start a word |
| 62 | 32100 | aleb | 0.000873 | |
| 63 | 18932 | Asked | 0.000873 | |
| 64 | 29756 | Isaiah | 0.000940 | Must start a word |
| 65 | 13084 | Survey | 0.000947 | Must start a word |
| 66 | 40944 | youngster | 0.000987 | Must start a word |

Table 7: Rejections of the manifold hypothesis for Llemma7B

| Number | Token ID | Token | $p$-value | Comment |
|---|---|---|---|---|
| 1 | 24812 | clojure | 0.00000000434 | |
| 2 | 10617 | agu | 0.000000846 | |
| 3 | 24391 | | 0.000000938 | Cyrillic; Must start a word |
| 4 | 6341 | custom | 0.00000163 | |
| 5 | 26226 | kW | 0.00000496 | Must start a word |
| 6 | 19579 | }^{- | 0.00000833 | |
| 7 | 17485 | porque | 0.0000128 | Must start a word |
| 8 | 23146 | Leopold | 0.0000135 | Must start a word |
| 9 | 5240 | Rem | 0.0000232 | Must start a word |
| 10 | 9317 | Ern | 0.0000265 | Must start a word |
| 11 | 2466 | though | 0.0000540 | Must start a word |
| 12 | 3413 | č | 0.0000619 | Must start a word |
| 13 | 18659 | endl | 0.0000683 | Must start a word |
| 14 | 24114 | corrected | 0.0000738 | Must start a word |
| 15 | 3252 | tw | 0.0000788 | Must start a word |
| 16 | 5908 | ères | 0.000115 | |
| 17 | 22385 | grep | 0.000151 | |
| 18 | 2924 | kind | 0.000202 | Must start a word |
| 19 | 16352 | pax | 0.000243 | |
| 20 | 8602 | Tri | 0.000250 | Must start a word |
| 21 | 18681 | spielte | 0.000258 | Must start a word |
| 22 | 22977 | Heaven | 0.000262 | Must start a word |
| 23 | 31946 | | 0.000289 | d with underline |
| 24 | 7949 | | 0.000320 | Cyrillic |
| 25 | 9160 | Card | 0.000335 | Must start a word |
| 26 | 14406 | Wall | 0.000351 | Must start a word |
| 27 | 29591 | periodic | 0.000359 | Must start a word |
| 28 | 10236 | ucht | 0.000463 | |
| 29 | 18308 | jeu | 0.000553 | Must start a word |
| 30 | 10029 | slightly | 0.000807 | Must start a word |
| 31 | 15935 | passion | 0.000823 | Must start a word |
| 32 | 5322 | Charles | 0.000833 | Must start a word |
| 33 | 15354 | victory | 0.000906 | Must start a word |

Table 8: Rejections of the manifold hypothesis for Mistral7B

| Number | Token ID | Token | $p$-value | Comment |
|---|---|---|---|---|
| 1 | 20639 | CV | 0.000000241 | Must start a word |
| 2 | 1314 | ember | 0.00000199 | |
| 3 | 31684 | | 0.00000224 | Korean |
| 4 | 31567 | ǐ | 0.00000770 | |
| 5 | 8418 | rise | 0.0000283 | Must start a word |
| 6 | 20508 | poured | 0.0000305 | Must start a word |
| 7 | 17939 | aimed | 0.0000364 | Must start a word |
| 8 | 21637 | allocate | 0.0000553 | Must start a word |
| 9 | 13215 | UINT | 0.0000684 | |
| 10 | 9009 | monitor | 0.0000767 | Must start a word |
| 11 | 25867 | | 0.000105 | Cyrillic |
| 12 | 30314 | \u0006 | 0.000132 | |
| 13 | 31517 | | 0.000133 | Chinese |
| 14 | 7402 | Lat | 0.000139 | Must start a word |
| 15 | 24351 | Vel | 0.000148 | |
| 16 | 10369 | challenges | 0.000178 | Must start a word |
| 17 | 30085 | | 0.000200 | Chinese |
| 18 | 11424 | Od | 0.000222 | Must start a word |
| 19 | 26983 | neys | 0.000244 | |
| 20 | 23851 | selon | 0.000244 | Must start a word |
| 21 | 10582 | measures | 0.000268 | Must start a word |
| 22 | 27460 | assumes | 0.000321 | Must start a word |
| 23 | 13542 | Et | 0.000338 | Must start a word |
| 24 | 13722 | Rot | 0.000438 | |
| 25 | 31097 | | 0.000538 | Chinese |
| 26 | 25244 | olutely | 0.000540 | |
| 27 | 4104 | HO | 0.000567 | |
| 28 | 24806 | Solutions | 0.000598 | Must start a word |
| 29 | 29587 | * | 0.000658 | |
| 30 | 12862 | Condition | 0.000665 | |
| 31 | 3903 | zt | 0.000677 | |
| 32 | 3903 | iced | 0.000696 | |
| 33 | 5343 | URL | 0.000722 | |
| 34 | 21518 | SEO | 0.000728 | Must start a word |
| 35 | 19929 | weigh | 0.000728 | Must start a word |
| 36 | 19929 | änge | 0.000765 | |
| 37 | 19929 | Vic | 0.000777 | Must start a word |
| 38 | 7566 | apparent | 0.000780 | Must start a word |
| 39 | 31780 | | 0.000896 | Chinese |
| 40 | 230 | <0xE3> | 0.000921 | |

Table 9: Rejections of the manifold hypothesis for Pythia6.9B (1 of 2)

| Number | Token ID | Token | $p$-value | Comment |
|---|---|---|---|---|
| 1 | 49503 | psychotic | 0.000000161 | Must start a word |
| 2 | 49503 | \\]( | 0.00000147 | |
| 3 | 13435 | 1990 | 0.00000231 | |
| 4 | 31918 | tiene | 0.00000282 | Must start a word |
| 5 | 12202 | ucky | 0.0000134 | |
| 6 | 25260 | Hebrew | 0.0000141 | Must start a word |
| 7 | 46631 | mess | 0.0000184 | |
| 8 | 5272 | reasonable | 0.0000300 | Must start a word |
| 9 | 25389 | collector | 0.0000311 | Must start a word |
| 10 | 24898 | embodiments | 0.0000428 | Must start a word |
| 11 | 47769 | Inhibition | 0.0000593 | Must start a word |
| 12 | 41264 | Hers | 0.0000691 | Must start a word |
| 13 | 13676 | odge | 0.000120 | |
| 14 | 49669 | | 0.000144 | Turkish |
| 15 | 16287 | | 0.000165 | Turkish |
| 16 | 26397 | 198 | 0.000170 | Must start a word |
| 17 | 30272 | scare | 0.000192 | Must start a word |
| 18 | 41305 | ]{}\\_[ | 0.000202 | |
| 19 | 25666 | LIMITED | 0.000212 | Must start a word |
| 20 | 5184 | hop | 0.000232 | Must start a word |
| 21 | 39253 | BIO | 0.000246 | |
| 22 | 48089 | raison | 0.000252 | Must start a word |
| 23 | 40084 | inside | 0.000254 | |
| 24 | 296 | st | 0.000261 | |
| 25 | 34475 | waving | 0.000286 | Must start a word |
| 26 | 21753 | Austria | 0.000287 | Must start a word |
| 27 | 5134 | ny | 0.000313 | |

Table 10: Rejections of the manifold hypothesis for Pythia6.9B (2 of 2)

| Number | Token ID | Token | $p$-value | Comment |
|---|---|---|---|---|
| 28 | 46728 | bags | 0.000326 | |
| 29 | 28672 | medium | 0.000372 | |
| 30 | 15210 | incent | 0.000378 | Must start a word |
| 31 | 21742 | carboh | 0.000407 | Must start a word |
| 32 | 4279 | secret | 0.000451 | Must start a word |
| 33 | 50231 | | 0.000468 | misc. Unicode |
| 34 | 40146 | | 0.000519 | Turkish, Must start a word |
| 35 | 8193 | 120 | 0.000536 | |
| 36 | 47334 | wastes | 0.000565 | Must start a word |
| 37 | 32232 | ressor | 0.000570 | |
| 38 | 22609 | Lip | 0.000597 | Must start a word |
| 39 | 42150 | preview | 0.000650 | |
| 40 | 19886 | redund | 0.000653 | Must start a word |
| 41 | 41962 | ridic | 0.000673 | Must start a word |
| 42 | 48444 | Ramirez | 0.000698 | Must start a word |
| 43 | 8977 | omatic | 0.000707 | |
| 44 | 46972 | Gloria | 0.000733 | Must start a word |
| 45 | 18807 | eton | 0.000780 | |
| 46 | 16782 | chairman | 0.000789 | Must start a word |
| 47 | 6413 | Supreme | 0.000859 | Must start a word |
| 48 | 42351 | grazing | 0.000890 | Must start a word |
| 49 | 5743 | activation | 0.000892 | Must start a word |
| 50 | 34282 | imming | 0.000914 | |
| 51 | 46513 | radiological | 0.000931 | Must start a word |
| 52 | 11363 | branc | 0.000941 | Must start a word |
| 53 | 47183 | $\\\\\\ | 0.000942 | Must start a word |
| 54 | 25409 | DEFAULT | 0.000992 | |

