# OpenReview forum: "Token Embeddings Violate the Manifold Hypothesis"
_NeurIPS.cc/2025/Conference — NeurIPS 2025 poster_

### Official Review · Reviewer_ZHkg · 2025-06-20

**Clarity:** 2
**Significance:** 2
**Originality:** 3
**Rating:** 3
**Confidence:** 4

**Summary:**

The manuscript  introduces an analysis aimed at verifying if the points of a dataset can be meaningfully considered as contained in a manifold or in a fiber bundle. The statistical test is applied for each points separately, and is based on a null model in which the manifold is locally flat and uniformly populated. The approach is applied to the analysis to token emebeddings in four large language models, and it allows identifying, for each model, between 40 and 70 tokens which violate the manifold hypothesis with high statistical confidence. A smaller number of tokens violates even the fiber bundle hypothesis.

**Questions:**

1) how does the intrinsic dimension  (ID) vary across the dataset?
2) how does the values of the ID obtained by a linear fit compare with those obtained by other ID estimators? For example you should consider the Danco Estimator, or the approach https://www.stat.berkeley.edu/~bickel/mldim.pdf
3) can you exclude that  the violations of the manifold hypothesis are due to variations of the density?
4) if one would eliminate from the vocabulary the (few) "singular" tokens, substituting those by synonyms, would the data manifold become everywhere smooth?

**Ethical Concerns:**

["NO or VERY MINOR ethics concerns only"]

**Final Justification:**

I changed my score from 2 to 3 since in my first review I did not notice that the value of the local dimension is reported in fig 2. My main remaining concern  is that the number of violations of the manifold hypothesis is really small. The authors use the lollypop as an example to explain why a small number of violations implies that a geometrical object is not a manifold. This is for sure correct, but, in my opinion, a bit off-topic in the contest of the analysis of representations in DNNs. Most (if not all) the approaches I am aware of do not assume that those representations  are GLOBALLY contained in a genuine manifold, as it would be intended in mathematical physics, and the fact that the intrinsic dimension is highly variable is well known.

**Limitations:**

yes

**Paper Formatting Concerns:**

The abstract, in my opinion, is not of a sufficient quality. The same sentence, "Failing to reject..." is repeated almost identical two times. The last sentence of the abstract is, for me, obscure, and its meaning was not clarified by reading the paper.

**Quality:**

2

**Strengths And Weaknesses:**

Strengths: focusing on the geometric properties of the data set in the neighborhood of each point separately is interesting and brings insight on the properties of each single token. Indeed, the tokens violating the manifold hypothesis are listed in Supp Mat, and one can attempt to interpret why these specific tokens correspond to "singularities"
Weaknesses:
1) (minor) the geometric criterion at the basis of the approach (eq 1) is very similar to the approaches used from the seventies onward to estimate the fractal dimension of a dataset (for example the correlation dimension). It is well known that one can obtain  deviations from  a linear behavior of the log-log plot of the volume versus radius just because the probability density of the data on the manifold varies quickly.
2) (major): the number of violations detected by the approach is very small, of the order of 0.1 % (according to table 1). Therefore, these results do not seriously challenge the idea that data representation are contained in a relatively low-dimensional manifold, also because, to the best of my knowledge,  nobody ever claimed that this manifold is a clean and smooth "Riemanian" variety. The fact that this manifold is not smooth everywhere is not a surprise.
3) (major): The first claim of section "Experimental results" is that  "the local dimension is highly variable".  Leaving aside the fact that even this result is not really novel, one  would like to see some numerical data supporting this claim.

---

> ### Author Rebuttal · Authors · 2025-07-31
>
> Thank you for your review for it made us realize that we had not been clear about the novelty and significance of the singular tokens.
>
> NB: Action items for the authors are marked with a "+" in the margin!
>
> We point out that even though they are quite few in number, singular tokens could be of extreme operational importance.  If we ignore/remove/substitute the singularities to "force" the token subspace to a manifold, we risk losing our ability to understand the behavioral features of the entire model.
>
> Indeed, after our present submission, this paper [Zhao et al, One Token to Fool LLM-as-a-Judge, arXiv:2507.08794, 2025] explains that some very specific tokens correspond to major vulnerabilities in LLMs.  These single-token attacks are becoming more widely known and prevalent [Cui et al, Token-Efficient Prompt Injection Attack: Provoking Cessation in LLM Reasoning via Adaptive Token Compression, arXiv:2504.20493, 2025], [Lakshmanan, "New TokenBreak Attack Bypasses AI Moderation with Single-Character Text Changes", published on The Hacker News website, June 12, 2025], [Zilberman, "Unicode Exploits Are Compromising Application Security", published on the Prompt.Security website, April 30, 2025].
>
> It is more than plausible that singularities might be the root cause of these vulnerabilities, and possibly the cause of other vulnerabilities not yet discovered. (That's future work!)
>
> +We will mention the above in our discussion Section 4.
>
> You are also correct that in a variety of settings, estimates of local dimension (which, depending on the author, has been called "intrinsic dimension"="ID", "local dimension", among various other terms) can vary quite a bit.  Also, you are correct that this variability has been reported by a number of others.  Our submission cites a few of these already [Fefferman 2018, Gromov 2024, Tulchinskii 2023, Robinson 2024], and the paper that you list is indeed another one of those.  Thanks for that!
>
> +We will include the paper you cite in our list of citations, mentioning it in Section 1.2
>
> +Also, since writing we found another good summary to mention in Section 1.2: [Schweinhart, Fractal dimension and the persistent homology of random geometric complexes, Advances in Mathematics, vol. 372, 2020]; we can include that paper in our citations to benefit the enterprising reader.
>
> Mathematically, a manifold has a singular dimension (pun intended), so the variability of dimension estimates is emblematic of manifold violation.  If the estimates of dimension were concentrated with a small variance around an integer, then we might conclude that the estimates were simply noisy.  Our results shown in Fig 2 and Table 1 show that this is not the case.
>
> Note that the variance is across all tokens for which intrinsic dimension can be estimated, which is a quite large number of them, so these estimates are reliable.  This is not in contradiction with the fact that for some of tokens, dimension cannot be estimated at all.
>
> So there are two ways our test has shown a violation of the manifold hypothesis. The large distribution of local dimension (as seen in the histogram in Fig 2 and Table 1) rejects that the token embedding is globally a manifold. This does not preclude that the embedding space is a disjoint union of smooth manifolds of different dimensions (which would still be problematic since LLMs leverage mathematical machinery such as inner products). Our test also rejects this possibility because we find tokens where a local dimension is simply not well defined. This implies something deeper like pinch points that link spaces of varied dimension. So in the lollipop example, the link point between the candy and the stick is singular because if we are at that point and look left, we appear to be in 1D while if we look right, we appear to be in 3D. The inner product that is being leveraged is on the extrinsic space and not the actual embedding space, which implies that the model is not operating on the token we would have wanted/expected. In fact, an inner product is not defined at this point (at least not in the usual way) and so we would have to characterize the singular point and find the correct mathematical machinery (if it exists).
>
> Moreover, in response to Q1, we do report the variability local dimension estimates in two places.  The histogram in the lower frame of Figure 2 shows the distribution of local dimension estimates for GPT-2.  Table 1 shows quartiles for the empirical distribution of local dimensions for each of the models we tested.  The fact that the interquartile range is large suggests that the local dimension estimates do indeed vary substantially.
>
> +We will rewrite Figure 2's caption to call out the fact that it is the distribution of dimension estimates, at least for the tokens whose dimensions can be estimated.
>
> Specifically, our primary/novel contribution is a *test*, not an *estimator*.  Our test asks whether there is a single reasonable estimate for dimension at each token.  If our test fails, then there is at least one token for which a single dimension does not make sense.  If our test fails, then any and all estimators (including the one that the referee cites) are not valid for use at that token, because the assumed relationship between volume and radius is not log-log linear.
>
> +We will insert the above two paragraphs into Section 1.1 so that there is no confusion
>
> There are a number of cases where the community implicitly assumes the data lies on a manifold, of the kind that formally satisfies the mathematical definition.
>
> +For avoidance of doubt, we will include the formal definition of a manifold in our paper, along with quite a few others that we mistakenly left out.  (See our response to Reviewer XXVi)
>
> Here are a few examples of when the manifold hypothesis is (strongly) assumed...using:
> 1. Inner products (which, in fact, requires a Riemannian manifold)
> 2. Cosine Similarity (which is just inner products on n-spheres which has even more assumptions about curvature especially negative curvature)
> 3. Manifold learning (Any sort of dimensionality reduction, etc)
>
> The key takeaway is that there are mathematical machinery for singular spaces...in practice, the community simply doesn't say the word "manifold" because we have always implicitly assumed we have a manifold, and often assumes a convex one at that. And so, an implication of our work is that there might be straightforward explanations when the model behaves erratically since we are using mathematical tools when we ought not. Without a Riemannian manifold as the embedding, it should not be surprising that applying inner products (loosely speaking) will generate unstable results.
>
> It is very much the case that *samples* drawn from a manifold under noisy conditions are also the norm.  That provides no issues for our results, as it is the formal setting of our paper, as noise is (explicitly!) incorporated into the volume form.  Indeed, this is why we use the standard hypothesis testing framework -- to handle noise in a statistically proper way.
>
> In the case of common manifolds, such as a sphere or torus, our manifold test never fails. In an earlier draft of the paper, we had included "toy" examples of our test being run on "noisy" manifolds and stratified manifolds.  However, we had to cut these examples out of the draft due to space constraints...
>
> +We will definitely include these examples (see below for some details) in our final version!
>
> The example of a stratified manifold is perhaps most important for the overall importance of our story.  As a simple example, consider a space that has the shape of an "X", like {(x,y) : |x| = |y|} in the 2D plane.  This space is not a manifold because it has exactly one singularity at the origin (0,0).  Local dimension at every other point is 1, but at the origin, the space does not have a well-defined dimension.  One can certainly remove the singularity to obtain a 1-dimensional manifold (with four connected components).
>
> Another example we plan to include is a "lollipop": a candy ball (hollow sphere) attached on the end of a short stick.  This shape has *exactly* two important singular points.  On the interior of the stick, the space is a 1-dimensional manifold.  At the very end of the stick, though, is a singular point.  Where the stick attaches to the "candy" part is another singularity; it is a transition from the 1-dimensional stick to the 3-dimensional candy part.
>
> However, removing the singularity, even though it accounts for 0\% of all points in the space, destroys the entire structure of the space!  The case of the token subspace is similar; even though the singularities constitute a vanishingly small number of tokens compared to the total vocabulary, they might be similarly important in interpreting the global structure of the token subspace.

---

> > ### Comment · Reviewer_ZHkg · 2025-08-04
> >
> > I thank the authors for their detailed reply. Concerning my question on the distribution of the local dimension, indeed the authors are right, the results were shown, and I apologise for not having noticed that. The most interesting feature of this probability distribution is that it is multimodal, with at least three modes, corresponding to different classes of tokens.  Since it was my mistake not noticing these results I raise my score to borderline reject (3).
> > I do not raise the score more  because the number of violations of the manifold hypothesis are too small to be relevant. This point has been also noticed by another reviewer. An example mentioned by the authors in their reply to explain why few violations of the manifold hypothesis are relevant  is a lollipop. Is a lollipop a manifold? The answer is clearly no. However, even on a lollypop  one can define a local tangent hyperplane almost everywhere, and therefore compute cosine similarities, euclidean distances, etc,  at least between nearby tokens.  Yes, on objects like this one should explicitly take into account that the intrinsic dimension is not constant. But also this fact has been addressed extensively in the literature, developing methods which allow finding if the data manifold can be decomposed in sub manifolds in which the ID is approximately constant.
> > Finally, the authors have not attempted to answer to my question 3 and 4. Question 3 can be easily answered by computing the local density, for example by the k-nn estimator, and checking if it correlates with the local dimension. Question 4 can be addressed by repeating the analysis on a subset of sentences which does not include the ~70 singular tokens.

---

> ### Author Response · Authors · 2025-08-04
> **Regarding Q4**
>
> Thank you for your response!
>
> Regarding Q4: This is a misunderstanding of what we did.  Our analysis is run on the *set* of tokens *only*.  No sentences (or token streams) were included.  Running "sentences" involves passing the token stream through the transformer blocks. That is upstream of the analysis we did.  Deleting the tokens from the model and then exploring its behavior is an interesting experiment, though is (a) future work and (b) not directly related to the experiments in our paper.
>
> Regarding Q3: We have not, but can for our revision, run this analysis.  In principle, this does not impact the results because of Theorem 1, but in practice we agree that it might have an effect due to sampling error.
>
> Finally and most importantly, about the "small number" of singularities:
>
> We agree: a lollipop is not a manifold!  Depending on exactly how it is defined, it could a stratified manifold.  It is the very fact that there are two singular points which implies that it is not a manifold.  And yes, a local dimension (which is not constant) can be defined almost everywhere.
>
> Yes, you can delete the two singular points, and obtain a disconnected, two component manifold.  However, we argue that the fact that the two components remain in close proximity to one another *is* significant, and moreover this is what our test finds.  Indeed -- it is unfortunate that we cannot include figures in our response -- in the figure of the results of our test run on said lollipop,
> 1. The exact points of singularity are not actually present, since the data are sampled, yet
> 2. You can see the p-values of our test decreasing in the neighborhood of these two singular points, with deep minima (reject the test!) right where those points are.  As a result, we get a handful of rejections near those singularities, depending on exactly what alpha you choose for significance.
>
> So in short, even if the singular points are deleted, their geometric presence still has a (statistically) significant impact on the token subspace!
>
> Moreover, the singular points are highly likely to have an impact on the behavior of the LLM as a whole, because according to our Theorem 2, they will propagate into the behavior of the model (ie., the transformer blocks that we did not study in this paper; see our response to Q4 above).

---

> ### Author Response · Authors · 2025-08-07
>
> Regarding the number of singular tokens, perhaps an (imperfect) analogy might be helpful in this discussion. The human genome has an estimated 20k-25k genes, and each gene encodes for one or more gene products. Across the population, mutations accumulate across and within each gene: some are innocuous, others might be beneficial, while many are deleterious. The statement that 70 irregular tokens do not meet a critical mass to warrant novelty/importance assumes that there is some sort of additive effect on irregular tokens that would be inconsequential on a per token level. This excludes the situation where even a single mutation/irregularity can have catastrophic consequences (for instance, a mutation to the BCRA1/2 gene increases the probability of cancer(s) by 60% to 90%). This also elicits the hub and spoke model from network theory where certain nodes are highly connected and extremely consequential.
>
> We believe that our work is complementary to the works that you have referenced (finding singular tokens contrasted with identifying regions of regularity / test vs estimator). Our work offers a new avenue of study for a previously overlooked structure (the irregular/singular tokens) to understand the magnitude of its impact on the global token embedding. We agree that there are ways to account for tokens with different intrinsic dimensions, but we stress that this is not currently implemented within the transformer models where the Q and K matrices return pairwise similarities based on normalized inner products (a structure that is being widely adopted into almost every new SOTA model). Our method further annotates a classification of tokens that do not have a well defined dimension which, to our knowledge, is not currently being done.
>
> We have obtained initial results showing the persistence of singular tokens within current transformers. We have, however, attempted to do some more exploratory data analysis on the singular tokens, and we will send an update to all of the reviewers in a subsequent note.
>
> Until we fully understand the nature of the singular token(s), we respectfully disagree with the statement that the number of irregularities "are too small to be relevant".

---

> ### Author Response · Authors · 2025-08-07
>
> In response to your (and another reviewers') concern over the scarcity of singular tokens, we attempted a quick and high-level analysis of these tokens relative to the other tokens. For this experiment, we performed an exploratory data analysis:
>
> We probed the Llemma-7B model with prompts consisting of a single token (preceded by the start token which is default for all Llemma prompts), as our work focuses entirely on the initial embedding and not the final embedding. This single token was varied over several cases:
> - Case 1: Llemma's singular tokens where the manifold hypothesis is rejected for small radii. Total = 21.
> - Case 2: Llemma's singular tokens where the manifold hypothesis is rejected for large radii. Total = 12.
> - Case 3: A random sample of Llemma's nonsingular tokens. We picked sample size = 33 to match the total number of singular tokens.
> - Case 4: All of Llemma's nonsingular tokens. Total = V - 33 = 31983, where V = 32016 is the vocabulary size.
>
> Since the sample in Case 3 is stochastic, we sampled 3 different times (we did not do more due to the time this takes to run over each token on our compute system; we cannot leverage cloud resources at the moment but could in the future).
>
> For each such prompt, we asked the model to generate output tokens under the greedy decoding scheme.
>
> Note that the internal layers of the model start from the token embeddings and eventually produce a probability distribution over the V = 32016 tokens at the final layer of the model. The token with the highest probability becomes the 'next' token that the model outputs. We conducted two analyses on these probabilities with a goal to quantify model behavior.
>
> In the first analysis, we collected the probability of the 'next' token. We aggregated these for all possible prompts in each Case outlined above, and looked at the maximum. The results are:
> - Case 1: 21 'small radii rejected' singular tokens: Max next token probability = 0.1268
> - Case 2: 12 'large radii rejected' singular tokens: Max next token probability = 0.7207
> - Case 3a: Sample of 33 nonsingular tokens: Max next token probability = 0.7852
> - Case 3b: Sample of 33 nonsingular tokens: Max next token probability = 0.934
> - Case 3c: Sample of 33 nonsingular tokens: Max next token probability = 0.8255
> - Case 4: All 31983 nonsingular tokens: Max next token probability = 0.9942
>
> The key takeaway from these results is that the maximum next token probability is only 0.1268 for Case 1. Thus, for ALL the singular tokens where the manifold hypothesis is rejected for small radii, when the model is prompted with them, it has very low confidence in its next token. One perspective is that it is very “confused” (inconclusive for the eyeball test for singular tokens from the large radius).
>
> In the second analysis, we computed the entropy of the 'next' token probability distribution (over V = 32016 outcomes) at the final layer. We aggregated these for all possible prompts in each case outlined above and looked at the minimum. The results are:
> - Case 1: 21 'small radii rejected' singular tokens: Min entropy of next token distribution = 5.6272
> - Case 2: 12 'large radii rejected' singular tokens: Min entropy of next token distribution = 2.16
> - Case 3a: Sample of 33 nonsingular tokens: Min entropy of next token distribution = 1.6605
> - Case 3b: Sample of 33 nonsingular tokens: Min entropy of next token distribution = 0.492
> - Case 3c: Sample of 33 nonsingular tokens: Min entropy of next token distribution = 1.2531
> - Case 4: All 31983 nonsingular tokens: Min entropy of next token distribution = 0.0598
>
> The key takeaway from these (quick and dirty) results is that the minimum entropy is as high as 5.6272 for Case 1, which is significantly higher than the nonsingular token cases. Thus, for ALL the singular tokens where the manifold hypothesis is rejected for small radii, when the model is prompted with them, its next token probability distribution is much closer to being uniform as compared to other cases where prompts contain nonsingular tokens (or singular tokens where the manifold hypothesis is rejected for large radii).
>
> We stress that this is a quick turnaround exploratory data analysis on these points. We could do this analysis across all of the LLMs in our draft, but we would prefer this to be future work so as to be able to design the experiments with more intentionality. We will leverage the advice from Reviewer TdY4 on embeddings studied in control settings that will allow us to calibrate these experiments. We don’t really think that there are any trivial/easy experiments since there are many levels of confounders that exist.

---

> ### Comment · Reviewer_ZHkg · 2025-08-08
> **Regarding Q4 (tokens or sequences) and the lollypop**
>
> Thanks very much for you detailed responses and for your extra work.
>
> Regarding Q4: "This is a misunderstanding of what we did. Our analysis is run on the set of tokens only. No sentences (or token streams) were included."
>
> The misunderstanding was generated by the following sentences in your manuscript: “Given these differences, prompts containing irregular tokens will likely produce dissimilar outputs across the four LLMs”
> “prompts containing irregular tokens within an LLM will likely be unstable and produce highly variable outputs with replicated queries.”
> “As we showed in Section 3.2, singularities in the token subspace persistently lead to singularities in the space of token sequences,”
> Prompts are token sequences, and you explicitly mention that the singularities you observe in the tokens propagate to token sequences. Even in the abstract you explicitly mention the effect of "singular" tokens on the stability of the reply to prompts. My question was aimed at understanding what changes by removing the few "singular" tokens. Even the last test you presented are based on next token prediction task, which implies the usage of a prompt, and you even look at what happens in the last layer (thus very far from the embedding space).
>
>  Citing again your manuscript, "In many—if not most—AI research papers, a manifold hypothesis is tacitly assumed, that the data are concentrated near a low curvature manifold without boundary." and you seem to provide evidence that this is not appropriate.  However, most of the analysis in the literature is performed in deep layers, and for representations generated by sentences, not single tokens. It is perfectly fine analysing the properties of the token embedding space as you are doing, but then you should be clear that the consequences of your analysis are restricted to this space. If you are convinced that singularities propagate in deep layers, you should provide a specific numerical evidence. With the results you have so far I think that the   sentences suggesting that the manifold hypothesis is inappropriate and the three sentences I mentioned above might create confusion in the reader, as they did in my case.
>
> Let me add one more time also my main point: in most of the non-linear analysis methods  I know (for example in Umap) it is not necessary assuming  that the data are contained in a manifold globally and at the large scale. Quoting my own first report "nobody ever claimed that this manifold is a clean and smooth "Riemanian" variety." Most non-linear  methods do work perfectly on the lollypop.

---

> > ### Author Response · Authors · 2025-08-08
> >
> > We do prove that the singularities propagate into the layers beyond the first, but we do not prove exactly what happens. This is an existence proof not a constructive proof. We know the existence of Nash equilibrium, just not how to find them. So what we say is technically true because of Theorem 2…but it can be somewhat unsatisfying as you are pointing out. We are certain that they propagate just as we are certain of fixed points called Nash Equilibrium for certain dynamical systems. We cannot provide numerical evidence currently without a great deal of computation which is not really the point. We can rephrase our paper to make our point much clearer…that our main result focuses on the initial embedding though we do provide mathematical guarantees on the later layers, and some of the statements we make are purely driven by Theorem 2.
> >
> > We are keenly aware of UMAP. In fact, our work is partially inspired by its formulation. To quote another (as it was easier to take their blog post verbatim):
> >
> > “There are also some assumptions made in justifying the formal mathematics that inspires the UMAP algorithm. The standard one, at least in machine learning, is the manifold hypothesis. The mathematical formalism in (Section 2, McInnes, Healy, and Melville 2018) makes some (potentially) stronger assumptions, namely that this manifold is paracompact and Hausdorff (in order to guarantee the existence of a Riemannian metric).”
> >
> > In fact, the authors of UMAP are saying exactly what you claim no one is saying concerning a Riemannian manifold. UMAP is built to in such a way that if your data clouds are highly heterogeneous (say a sample from a stratified manifold), it will obliterate the global structure in order to estimate local ones. This is a big concern and why mathematicians have argued against using UMAP as an analytic tool. Unless you have a (stronger) manifold hypothesis assumption, it can be wildly misleading.
> >
> > We agree that these non-linear methods seem to work some times (maybe even much of the times), but that is the danger because we won’t be able to differentiate when it is working and when it does not. Mathematicians prove theories to provide guarantees 100 percent of the times (constrained by assumptions), and so even a single counter-example violates the notion of a mathematical theorem. So from our point of view, we only needed 1 but provided around 70 such examples. This seems like domain cross-talk between us, and we will do more to clarify to a broader audience.

---

### Official Review · Reviewer_TdY4 · 2025-06-27

**Clarity:** 1
**Significance:** 3
**Originality:** 4
**Rating:** 4
**Confidence:** 2

**Summary:**

The authors propose three hypothesis tests designed to assess the geometric structure of embedding subspaces associated with individual tokens. Specifically, they aim to determine whether a subspace can be characterized as (1) a manifold, (2) a fibered manifold, or (3) a fiber bundle. These tests are grounded in Theorem 1, which describes how the volume of a ball of radius $r$, centered at a point $\psi$, changes for small versus large $r$. The authors apply their methodology to analyze token embeddings produced by large language models (LLMs).

**Questions:**

# Major Comments

**1. Presentation and Organization**
The paper lacks clear structure, which significantly hinders readability. For example, Figure 3 is said to encapsulate the core methodology, yet its interpretation is non-trivial without additional context. The necessary intuition is relegated to the appendix, where it is unlikely to be seen early by readers. I strongly recommend relocating this explanation to the main text and integrating it into the broader narrative.

**2. Volume Estimation in Figure 3b**
It is unclear how volume is defined or computed in Figure 3b. Lines 120–125 suggest that it may correspond to the number of tokens within a ball. If this is correct, the authors appear to be using token count as a proxy for manifold volume. This is a strong and potentially unjustified assumption. Can the authors provide a theoretical or empirical rationale for this estimation method?

**3. Slope Thresholding**
The hypothesis tests seem to hinge on detecting slope changes in volume scaling. Does this involve setting a specific threshold? If so, how was this threshold determined? Why was this particular value chosen over others? The paper should clarify the methodology for threshold selection and discuss its sensitivity.

**4. Experiments on Specific Token Subsets**
The authors’ conclusion---that LLM embeddings do not lie on a single coherent manifold---is intriguing, but more granular analysis is needed. For example, are certain semantic subsets (e.g., years from 1900–2000, color names, or month names) manifold-like? Exploring such structured subsets could strengthen the argument and help determine whether the negative results reflect global structure or just aggregate noise.

**5. Comparison to Other Embedding Methods**
The empirical evaluation is limited to LLM embeddings. However, other widely used embedding techniques, such as Word2Vec and GloVe, also produce token representations. Applying the proposed tests to these methods could provide important comparative insights and help clarify whether the results are specific to LLMs or generalizable across embedding strategies.

**6. Test Validity and Robustness**
The paper lacks validation of the proposed tests on synthetic data. For example, if the authors apply their methodology to points sampled from known manifolds (e.g., spheres, tori), do the tests recover the expected structure? How do noise and sampling density affect test accuracy? Exploring such controlled settings would provide valuable evidence for the tests’ robustness and reliability.

---

# Minor Comments

* **Equation Formatting**: Equation (1) contains an unclosed parenthesis: $(e_*v)(B_r(\phi)\color{red})\color{black}=\dots$

**Ethical Concerns:**

["NO or VERY MINOR ethics concerns only"]

**Final Justification:**

The authors have addressed my main concern by outlining a clear and reasonable revision plan. Additionally, they have provided satisfactory responses to my other comments. I have decided to raise my score.

**Limitations:**

Yes

**Paper Formatting Concerns:**

No concerns

**Quality:**

1

**Strengths And Weaknesses:**

## Strengths

* The paper addresses a timely and relevant problem.
* The theoretical framework appears sound and well-motivated.
* The conclusions challenge prevailing assumptions in the community.

## Weaknesses

* The paper's presentation is disorganized and often unclear.
* The experimental section lacks detail and depth.
* The empirical analysis is limited in scope.

## Review Summary

This work introduces a novel and theoretically grounded approach for testing the manifold hypothesis in token embeddings derived from LLMs. The conclusions are intriguing and potentially impactful, as they challenge commonly held beliefs about the geometry of learned representations.

However, the paper suffers from several major weaknesses. The organization and clarity of presentation are lacking---key intuitions and methodological explanations are buried in the appendix rather than integrated into the main narrative. Moreover, the experimental validation is limited both in scope and detail. The authors focus exclusively on LLM embeddings, ignoring other widely used embedding techniques such as Word2Vec or GloVe. Furthermore, while the proposed tests appear general, their empirical application is shallow, raising concerns about their validity and robustness.

Despite the paper’s potential, I do not recommend it for publication in its current form. I suggest a **borderline reject**.

That said, I acknowledge that I may have missed or misunderstood certain aspects of the paper, and my confidence in this assessment is low. I would be open to revising my recommendation if the authors are able to address the concerns outlined below.

Regardless of the final decision, I can see the potential of this work and I encourage the authors to continue their research in this direction.

---

> ### Author Rebuttal · Authors · 2025-07-31
>
> Thank you for your review and positive encouragement for the findings in our work.
>
> NB: Action items for the authors are marked with a "+" in the margin!
>
> > That said, I acknowledge that I may have missed or misunderstood certain aspects of the paper, and my confidence in this assessment is low. I would be open to revising my recommendation if the authors are able to address the concerns outlined below.
>
> Let us lead off by saying that we are happy to engage in discussion!
>
> Mathematically, a manifold has a singular dimension (pun intended), so the variability of dimension estimates is emblematic of manifold violation.  If the estimates of dimension were concentrated with a small variance around an integer, then we might conclude that the estimates were simply noisy.  Our results shown in Fig 2 and Table 1 show that this is not the case.
>
> Note that the variance is across all tokens for which intrinsic dimension can be estimated, which is a quite large number of them, so these estimates are reliable.  This is not in contradiction with the fact that for some of tokens, dimension cannot be estimated at all.
>
> So there are two ways our test has shown a violation of the manifold hypothesis. The large distribution of local dimension (as seen in the histogram in Fig 2 and Table 1) rejects that the token embedding is globally a manifold. This does not preclude that the embedding space is a disjoint union of smooth manifolds of different dimensions (which would still be problematic since LLMs leverage mathematical machinery such as inner products). Our test also rejects this possibility because we find tokens where a local dimension is simply not well defined. This implies something deeper like pinch points that link spaces of varied dimension. Imagine glueing a line segment to a ball (lollipop); the link point does not have a well-defined dim because it joins 1D to 3D. The inner product that is being leveraged at this link is on the extrinsic space and not the actual embedding space, which implies that the model is not operating on the token we would have wanted/expected. In fact, an inner product is not defined at this point (at least not in the usual way) and so we would have to characterize the singular point and find the correct mathematical machinery (if it exists).
>
> > The organization and clarity of presentation are lacking
>
> +We can bring material from A.3 and A.4 up into the main body Sec 2 without too much effort, especially because the final version permits an extra page.  (Do also have a look at our response to reviewer MkzF, wherein we have committed to a detailed revision plan.)
>
> > For example, Fig 3 is said to encapsulate the core methodology, yet its interpretation is non-trivial without additional context. The necessary intuition is relegated to the appendix, where it is unlikely to be seen early by readers. I strongly recommend relocating this explanation to the main text and integrating it into the broader narrative.
>
> That is a great idea.  Thanks!  We will also incorporate some text like what we say below (concrete actions marked)...
>
> > Volume Estimation in Fig 3b It is unclear how volume is defined or computed in Fig 3b. Lines 120–125 suggest that it may correspond to the number of tokens within a ball. If this is correct, the authors appear to be using token count as a proxy for manifold volume. This is a strong and potentially unjustified assumption.
>
> We are treating the tokens as a sample from the assumed space (either a manifold or a fiber bundle, depending on the test). We are estimating the volume of the ball of a given radius around a given token by counting the number of tokens within that radius. This is nothing other than a classic Monte Carlo estimate of volume. It is a strong assumption, and is inherent in the "volume form" in the statements of our Theorem 1, but it is also extremely established in the literature.
>
> +We will include the above paragraph (suitably edited) in Sec 2
>
> > Slope Thresholding The hypothesis tests seem to hinge on detecting slope changes in volume scaling. Does this involve setting a specific threshold? If so, how was this threshold determined? Why was this particular value chosen over others? The paper should clarify the methodology for threshold selection and discuss its sensitivity.
>
> Thank you for your careful read of the paper!
>
> For the manifold test, which is a little simpler, we collected dimension estimates using the standard 3-point centered difference estimate via `numpy.gradient()` at each radius under test.  These dimension estimates were then binned into a pair of adjacent sliding windows.  The estimates in neighboring bins were compared using the standard unpaired two-sample T-test with the stated alpha level for decision.  This test yields a Bonferroni-corrected p-value for the token.  If the p value is less than alpha, then there is not a well-defined dimension for that token, so the manifold test fails at that token.
>
> +We will include the above paragraph (suitably edited) in Sec 2
>
> > The authors’ conclusion---that LLM embeddings do not lie on a single coherent manifold---is intriguing, but more granular analysis is needed.
>
> That is a fascinating idea, but really ought to be future work.  We need to systematically collect data to conduct such an experiment.  Thank you for the suggestion!
>
> +We can list your suggestion as future work in Sec 4.
>
> > Exploring such structured subsets could strengthen the argument and help determine whether the negative results reflect global structure or just aggregate noise.
>
> They are not multiple runs, as the entire process in our paper is deterministic.  What we are presenting is an analytic tool for pre-trained models, that moreover can be applied to any point cloud within a latent space.  In other words, the token embedding vectors are fixed when we analyze the model, they do not change from run to run.
>
> A natural question is therefore if we can find a token embedding that is everywhere regular. One way to approach an answer is to view the number of tokens in a vocabulary as a hyperparameter.  There might be a certain vocabulary size that minimizes the number of singularities given the corpus, architecture, etc.  Even the best vocabulary might lead to a space that could still be singular, for reasons inspired by [Jakubowski et al, Topology of Word Embeddings: Singularities Reflect Polysemy, in Proceedings of the Ninth Joint Conference on Lexical and Computational Semantics, 2020].  Let's take the * symbol, which happens to be singular for Mistral 7B (See Table 7, number 29). In natural language, it has a very distinctive meaning in terms of calling out footnotes, while in other contexts it can be used for marking text as bold in Markdown, or in programming it has the mathematical meaning of multiplication.  Therefore, this character connects these different semantics: i.e. natural language with programming languages and markdown.  Suppose that each language were in fact well-described by a manifold.  Then these manifolds would be joined together in the entire latent space at the * (and likely other points).  As a result, this token with high likelihood would be a singularity because the dimensions for different "languages" are almost surely different [Tulchinskii et al, Intrinsic Dimension Estimation for Robust Detection of AI-Generated Texts, arxiv:2306.04723, 2023].
>
> +We can include the above discussion in the introduction (space permitting) or Discussion Sec 4.
>
> +We will also mention something about your question about randomness in our Limitations Sec 4.1 (since although it is not a limitation, the Limitations section is where readers with similar questions might look.)
>
> > The empirical evaluation is limited to LLM embeddings. However, other widely used embedding techniques, such as Word2Vec and GloVe, also produce token representations. Applying the proposed tests to these methods could provide important comparative insights and help clarify whether the results are specific to LLMs or generalizable across embedding strategies.
>
> This is a fair point, and it's entirely sensible to run our test(s) on these other embeddings.  The method applies to any point cloud in Euclidean space, and LLMs are mostly a convenient test case. We opted for LLM embeddings because they are currently the most widely deployed models where we could easily make some high level comparisons. Comparing across model embeddings is complicated by potential confounding effects, e.g. corpus, cost function, etc. This is certainly future work.
>
> > The paper lacks validation of the proposed tests on synthetic data. For example, if the authors apply their methodology to points sampled from known manifolds (e.g., spheres, tori), do the tests recover the expected structure? How do noise and sampling density affect test accuracy? Exploring such controlled settings would provide valuable evidence for the tests’ robustness and reliability.
>
> Regarding a synthetic example, in an earlier draft of the paper, we had included "toy" examples of our test being run on "noisy" manifolds and stratified manifolds.  However, we had to cut these examples out of the submission due to space constraints...
>
> +we will include these examples in our final version!
>
> Since we are not permitted to include graphics in our rebuttals, for the purpose of visualization, one example we plan to include is the "lollipop": a candy ball attached on the end of a short stick (mentioned before).  This shape has EXACTLY two singular points.  On the interior of stick, the space is a 1-dimensional manifold.  The end of the stick is a singular point/boundary.  Where the stick attaches to the "candy" is another singularity; it is a transition from the 1-dimensional stick to the 3-dimensional candy.
>
> > Equation Formatting: Equation (1) contains an unclosed parenthesis:
>
> +Oops.  Thanks for catching that!

---

> > ### Comment · Reviewer_TdY4 · 2025-08-02
> >
> > I find merit in this submission and acknowledge the value of the work presented. I trust the authors will thoughtfully incorporate the points addressed in their rebuttal. Accordingly, I am raising my score.

---

> > > ### Comment · Reviewer_TdY4 · 2025-08-05
> > >
> > > This paper presents an interesting idea and, in my view, explores a promising direction. That said, the manuscript currently feels a bit rough around the edges.
> > >
> > > Based on the author response, I have decided to raise my score. I believe the core idea is stronger than a borderline accept, but the current presentation of the manuscript limits how much higher I can go.
> > >
> > > For the next revision, I encourage the authors to better convey the intuition and technical details of their approach—ideally through visual or toy examples that can help ground the reader.
> > >
> > > I also recommend that the authors explore the literature on grokking, where embeddings are studied in more controlled settings. This line of work could inform more focused experiments, for instance on the effects of different optimizers, weight decay, data noise, and other factors.
> > >
> > > Regardless of the outcome of this process, I wish the authors the best of luck with their work.

---

> ### Author Response · Authors · 2025-08-08
>
> Thank you once again for your review and additional engagement with our response, we appreciate it.
>
> Additionally, please note that in response to some reviewers' comments about the number of singular tokens, we attempted a quick exploratory data analysis of singular tokens relative to the other tokens. In case you are interested to see the results, they are in our responses (i.e. official comments) to Reviewers ZHkg and XXVi.

---

### Official Review · Reviewer_XXVi · 2025-07-01

**Clarity:** 1
**Significance:** 2
**Originality:** 4
**Rating:** 3
**Confidence:** 3

**Summary:**

This work investigates the presence of irregularities and singularities in the structure of token embedding spaces used by large language models. Treating the embedding space as a geometric object, the authors evaluate its conformity to the manifold hypothesis and the more general fiber bundle structure. They introduce statistical tests to identify tokens whose embeddings violate these hypotheses, revealing cases where the space is locally inconsistent or singularities. The authors also provide a theoretical justification for why such singularities can persist through the model, even with extended context, potentially affecting downstream behavior. Finally, they analyze patterns of geometric irregularity across multiple models, identifying patterns among tokens which violate the hypothesis.

**Questions:**

Q1: Could the authors point to works in literature which previously suggested adding more tokens to the context help reduce singularities ?

Q2: Could you please elaborate on the significance of Fig 2 bottom ?

Q3: There are a number of works which have discussed a souce of potential geometric similarities across models to be due to shared dataset and next token distributions. Some references are Lee et al  2024 (Shared Global and Local Geometry of Language Model Embeddings) and  Zhao et. al 2024 (Impliciteometry of Next Token Prediction) . Do the authors have any thoughts on this, specially regarding how some of the singular tokens are shared among models such as Mistral and GPT ? Could this all be an artifact of the training data and the next token distribution ?

**Ethical Concerns:**

["NO or VERY MINOR ethics concerns only"]

**Final Justification:**

Apologies for submitting this late, I thought it was only for cases where the score changes. I see the merit in the work and I understand its potential. The authors do a great job at responding to some of the points raised by other reviewer and myself. In particular I believe their final point regarding the impact of the singular tokens on ntp probability is quite impressive. I believe the work is acceptable but I believe that the work requires more clarity in its presentation and needs to include some of the points discussed during this review process for me to raise my score. Therefore, I am keeping my original rating.

**Limitations:**

Addressed

**Quality:**

2

**Strengths And Weaknesses:**

Strengths
The paper presents an original and novel perspective on the geometry of token embedding spaces in large language models. By framing the embedding space through the lens of the manifold hypothesis and evaluating deviations via both manifold and fiber bundle tests, the authors introduce a methodology for identifying structural irregularities in token representations. This line of inquiry opens up potential avenues for systematically characterizing LLM embedding geometry and I appreciate the use of empirical analysis in topological and geometric theory is commendable.

Weaknesses
While the contribution is conceptually intriguing, the paper suffers from several issues that limit accessibility and clarity. Many of the key mathematical terms — such as manifold, fiber bundle, and singularity — are introduced without sufficient formal definition or intuitive explanation. This makes it difficult for readers unfamiliar with differential geometry (such as myself) to follow the central arguments. Additionally, some theoretical results, particularly Theorem 1 and Theorem 2, are introduced and referenced before they are clearly stated.

From a presentation standpoint, several figures are difficult to interpret. I'm having a hard time understanding the significance of Fig1 and Fig2 bottom, the flowcharts in Fig2.

Finally, a central concern is the scale and significance of the results. If interpreted table1 correctly, fewer than 70 tokens (out of ~30K–50K vocabulary) fail the manifold hypothesis test, and the number of fiber bundle violations, while higher, still represents a relatively small fraction (again if I understand table 1 correctly). While the authors argue these irregularities are meaningful, it's unclear whether these patterns are consistent across training runs or simply artifacts of randomness in high-dimensional embedding spaces. This raises questions about the practical implications and generality of the findings.

I would like to continue this conversation to clarify some of the points in the rebuttal phase.

---

> ### Author Rebuttal · Authors · 2025-07-31
>
> Thank you for your review and pointing out that we were not clear on the novelty and significance of the singular tokens.
>
> NB: Action items for the authors are marked with a "+" in the margin!
>
> > I would like to continue this conversation to clarify some of the points in the rebuttal phase.
>
> Great!  We are delighted to discuss this with you!  We appreciate your effort in reading our paper.
>
> +We will define in Section 1.2
>
> * manifold
> * singularity (and separately "singular")
> * embedding
> * embedded reach
> * volume form
> * radius
> * fibered manifold
> * fiber bundle
> * p-value
>
> +In the appendix
>
> * curvature
> * pushforward
> * homeomorphism
> * diffeomorphism
>
> > Additionally, some theoretical results, particularly Thm 1 and Thm 2, are introduced and referenced before they are clearly stated.
>
> +We will move up some of the material from A.3 and A.4 into Sec 2 so that that is rather more clear.  (see also the response to Reviewer TdY4)
>
> >If interpreted table1 correctly, fewer than 70 tokens (out of ~30K–50K vocabulary) ...
>
> We point out that even though they are quite few in number, these singular tokens could be of extreme operational importance.  Individual tokens are already known to be the root cause of certain vulnerabilities.  For instance, [arXiv:2507.08794] explains that some very specific tokens correspond to major vulnerabilities in LLMs (called the "master-key" in unlocking LLM prompt injections).  These single-token attacks are becoming more widely known and prevalent [arXiv:2504.20493], [Lakshmanan, "New TokenBreak Attack Bypasses AI Moderation with Single-Character Text Changes", 2025], [Zilberman, "Unicode Exploits Are Compromising Application Security", 2025].
>
> It is plausible that singularities might be the root cause of these vulnerabilies.  Moreover, these singular tokens might be the cause of other vulnerabilities not yet discovered. (That's future work!)   We would argue that since the singular tokens are distinctly different than the others, and that there are not that many of them, finding the singular tokens is potentially of great value to be community.
>
> +We will insert the above text (after editing) into the Introduction Sec 1 or the Discussion Sec 4.
>
> > While the authors argue these irregularities are meaningful, it's unclear whether these patterns are consistent across training runs
>
> Note that the entire process in our paper is deterministic, so there is no requirement of multiple runs.  What we are presenting is an analytic tool (hypothesis test) for pre-trained models, that moreover can be applied to any point cloud within a latent space.  Note that the models we are considering are published/deployed models, where the token embedding vectors are frozen after training. In other words, the token embedding vectors are fixed when we analyze the model, they do not change from run to run.
>
> +We will add the above text (suitably edited) into the Background Sec 1.2, and reiterate it in our Limitations Sec 4.1 (since although it is not a limitation, the Limitations section is where readers with similar questions might look.)
>
> > Q1
>
> We have actually never found this discussed in the literature at all.  Indeed, we found (and already cited) only a few papers that mention singularities at all.  What we are saying is that we suspect one of the motivations for the trend toward increasing context window size might be a response to the presence of singularities.
>
> We also stress that our work shows that if the initial token embedding contains singular tokens, they will persist in the final embedding regardless of the number of tokens (i.e. the context window size) by Thm 2. Our result is not a contradiction of prior work but simply a result that stands on its own.
>
> +Assuming the above paragraph clarifies your question, we could add it to the Discussion Sec 4.
>
> > Q2
>
> +Fig 2 bottom is the histogram of the estimates of dimensions for GPT2 (=slopes of the log radius versus log volume plots), at least for when the tokens have a well-defined dimension.
>
> +We will update the caption of Fig 2 to say the above explicitly.
>
> So there are two ways our test has shown a violation of the manifold hypothesis. The large distribution of local dimension (as seen in the histogram in Fig 2 and Table 1) rejects that the token embedding is globally a manifold. This does not preclude that the embedding space is a disjoint union of smooth manifolds of different dimensions (which would still be problematic since LLMs leverage mathematical machinery such as inner products). Our test also rejects this possibility because we find tokens where a local dimension is simply not well defined (these are our highlighted singular tokens). This implies something deeper like pinch points that link spaces of varied dimension. Imagine glueing a line segment to a ball (lollipop); the link point does not have a well-defined dim because it joins 1D to 3D. The inner product that is being leveraged at this link is on the extrinsic space and not the actual embedding space, which implies that the model is not operating on the token we would have wanted/expected.  In fact, an inner product is not defined at this point (at least not in the usual way) and so we would have to characterize the singular point and find the correct mathematical machinery (if it exists).
>
> > Q3
>
> While there might be good alignment between models over much of the token subspace (which would account for the broad geometric similarities), there *simply cannot be* alignment between the singular tokens *if the singular tokens differ* or if the dimensions differ.  Since we show (Tables 3-9) that the singular tokens between the four models we studied differ, there *cannot* be alignment between these models in a fully global sense.
>
> > Some references are Lee et al 2024 (Shared Global and Local Geometry of Language Model Embeddings)
>
> The authors of this paper show a few things that are consistent with our findings:
>
> +We will discuss this paper explicitly in Sec 1.2, as below:
>
> 1. Language models often construct similar local representations as constructed from local linear embeddings
>
> The "often" part here is important.  Our finding is that most of the tokens in the vocabulary fail to reject the manifold hypothesis, which means that we are able to estimate some local structure.  However, for the singular tokens this is simply impossible without deep investigation of the local neighborhoods.  In [Lee 2024], they are not looking at the singular tokens in the way that we are here.
>
> 2. Token embeddings exhibit low intrinsic dimensions
>
> This agrees with our findings in Table 1, at least in the lower radius regime.  Their estimates of dimension for GPT2 is within the confidence interval we computed.  They do not recognize that there might be a fiber bundle structure in play, which is our "larger radius regime".
>
> 3. Tokens with lower intrinsic dimensions form more semantically coherent clusters.
>
> We agree, though since they did not perform a test for whether a given token is singular, we are concerned that they may have estimated dimensions for tokens that do not have a well-defined dimension!
>
> Nevertheless, Figure 1 seems fairly consistent with their findings.
>
> 4. Tokens have similar dimensions across language models
>
> The hypothesis that "Tokens have similar dimensions across language models" is a null hypothesis.  It can be rejected (simply by exhibit two models whose dimensions disagree), but never accepted as true since *all* models cannot be tested.  Our Table 1 shows that this hypothesis must be rejected.  It is not that the dimensions estimated by [Lee 2024] are wrong.  We and they both looked at four models (with one in common, GPT2).  It is highly likely that they might have--by chance alone--picked models that are similar to one another and thereby did not find that their dimensions could differ.
>
> > Zhao et. al 2024 (Implicit geometry of Next Token Prediction)
>
> This paper is about the transformer structure (our Section 3.2), but not so much about the token embeddings explicitly, so it is less directly relevant to our present submission, but see below.
>
> +We will include the following two paragraphs (perhaps summarized after discussion with you) in Section 1.2.
>
> Specifically, they show that the implicit space of possible activations can be inferred from the behavior of the model.  A *slice* of this implicit structure is the token subspace.  We note that in this vein, [Zhao 2024] is not inconsistent with other findings within the literature, for instance [Robinson et al, Probing the topology of the space of tokens with structured prompts, arXiv:2503.15421, 2025].  This latter paper shows that the token subspace can be inferred (up to homeomorphism, aka topological equivalence) from the responses of the LLM to single-token prompts.
>
> Taken together, the above papers and the present one have pretty significant implications.  Even though the token subspace may have about 0.1% of tokens that are singular, we suspect that a user will be encountering singular tokens on a regular basis when using LLMs (which seems empirically plausible).  Estimating the rate of occurance of a singular token is hard, and very much future work!
>
> However, this matters in two distinct ways:
> 1. Prompting from the user.  If the user hits a singular token, both [Zhao 2024] and [Robinson 2025] indicate that the model's behavior is likely to go badly from that point onward.  The current "fix" for this problem is prompt engineering.
>
> 2. As part of how they work, the transformers sample the next token, and almost surely as the number of samples gets very large, the LLM will indeed pick a singular token. There is no prompt engineering fix in this case.
>
> > Could this all be an artifact of the training data and the next token distribution ?
>
> We are not entirely sure about the specific meaning of your question.  Please clarify?

---

> ### Author Response · Authors · 2025-08-07
>
> Since the official response period has been extended by two days, there is no apologies needed for not replying until now. We are obliged for your feedback.
>
> Thank you for the clarification on the artifacts of randomness. There juxtaposition of “artifact” with “randomness” confused us as work in error modeling can usually be decomposed into random noise (stochastic errors) or artifacts (systematic bias). While we have not done any principled analysis into this questions, we can do some thought experiments to inform our intuition. If we think about training an LLM as an experiment, then the experimental design/setup/conditions severely biases the outcomes with artifacts. These systematic effects are unique to each experiment; in fact, they are unique to batches of the experiment (the batch correction problem severely limits the analysis of molecular data). This would certainly suggest different training corpuses, architecture, etc. would drive systemic differences between models, and token embeddings would also likely be biased. Stochasticity, on the other hand, is a bit more tricky to intuit. If we could potentially fix all other aspects of training, there is still a stochastic component to the non-convex optimization -- the solution to which you obtain for any given training run. One could imagine that embeddings are influenced by the local geometry of the zero set of the cost function. In reality, the experimental design (architecture, corpus, cost function, hyperparameter choices, regularizers, etc.) influences/determines the loss landscape. So yes, we do think that the nature of the experiment would impact the geometry of the token embeddings where both the fixed effects (experimental design) as well as the random effects (stochastic variability) play a role in the learning of an irregular/singular embedding space.
>
>
> As for question 1, our work shows that adding more tokens cannot help to reduce singularities if they are already present (Theorem 2). This means that the singularities will persist across the network and into the final embedding regardless of adding more tokens (or equivalently, adding context).
>
> This histogram in Figure 2 shows the distribution of the per token estimate for the intrinsic dimension within the token embedding. It shows that the local dimension for each token (when it is well defined) varies considerably. This variability itself violates the manifold hypothesis as it implies that the token embedding does not have a unique dimension (which a manifold must have by definition). The singular tokens are not represented in this histogram as they don’t have a well defined dimension. For instance, we can take a line segment and glue this to a solid sphere. The junction point does not have a well defined local dimension because on one side it is 1D but on the other side, it is 3D (so we cannot define local charts, etc). This is the lollipop stratified manifold toy example.

---

> ### Author Response · Authors · 2025-08-07
>
> We did not present evidence in our submission that the singular tokens would be problematic as ours is a methods paper and not an analysis paper. Because of your (and others') question on the nature of the singular tokens, we conducted some experiments in an attempt to understand this better in the past few days (we are sending this note to all reviewers):
>
> In response to your (and another reviewers') concern over the scarcity of singular tokens, we attempted a quick and high-level analysis of these tokens relative to the other tokens. For this experiment, we performed an exploratory data analysis:
>
> We probed the Llemma-7B model with prompts consisting of a single token (preceded by the start token which is default for all Llemma prompts), as our work focuses entirely on the initial embedding and not the final embedding. This single token was varied over several cases:
> - Case 1: Llemma's singular tokens where the manifold hypothesis is rejected for small radii. Total = 21.
> - Case 2: Llemma's singular tokens where the manifold hypothesis is rejected for large radii. Total = 12.
> - Case 3: A random sample of Llemma's nonsingular tokens. We picked sample size = 33 to match the total number of singular tokens.
> - Case 4: All of Llemma's nonsingular tokens. Total = V - 33 = 31983, where V = 32016 is the vocabulary size.
>
> Since the sample in Case 3 is stochastic, we sampled 3 different times (we did not do more due to the time this takes to run over each token on our compute system; we cannot leverage cloud resources at the moment but could in the future).
>
> For each such prompt, we asked the model to generate output tokens under the greedy decoding scheme.
>
> Note that the internal layers of the model start from the token embeddings and eventually produce a probability distribution over the V = 32016 tokens at the final layer of the model. The token with the highest probability becomes the 'next' token that the model outputs. We conducted two analyses on these probabilities with a goal to quantify model behavior.
>
> In the first analysis, we collected the probability of the 'next' token. We aggregated these for all possible prompts in each Case outlined above, and looked at the maximum. The results are:
> - Case 1: 21 'small radii rejected' singular tokens: Max next token probability = 0.1268
> - Case 2: 12 'large radii rejected' singular tokens: Max next token probability = 0.7207
> - Case 3a: Sample of 33 nonsingular tokens: Max next token probability = 0.7852
> - Case 3b: Sample of 33 nonsingular tokens: Max next token probability = 0.934
> - Case 3c: Sample of 33 nonsingular tokens: Max next token probability = 0.8255
> - Case 4: All 31983 nonsingular tokens: Max next token probability = 0.9942
>
> The key takeaway from these results is that the maximum next token probability is only 0.1268 for Case 1. Thus, for ALL the singular tokens where the manifold hypothesis is rejected for small radii, when the model is prompted with them, it has very low confidence in its next token. One perspective is that it is very “confused” (inconclusive for the eyeball test for singular tokens from the large radius).
>
> In the second analysis, we computed the entropy of the 'next' token probability distribution (over V = 32016 outcomes) at the final layer. We aggregated these for all possible prompts in each case outlined above and looked at the minimum. The results are:
> - Case 1: 21 'small radii rejected' singular tokens: Min entropy of next token distribution = 5.6272
> - Case 2: 12 'large radii rejected' singular tokens: Min entropy of next token distribution = 2.16
> - Case 3a: Sample of 33 nonsingular tokens: Min entropy of next token distribution = 1.6605
> - Case 3b: Sample of 33 nonsingular tokens: Min entropy of next token distribution = 0.492
> - Case 3c: Sample of 33 nonsingular tokens: Min entropy of next token distribution = 1.2531
> - Case 4: All 31983 nonsingular tokens: Min entropy of next token distribution = 0.0598
>
> The key takeaway from these (quick and dirty) results is that the minimum entropy is as high as 5.6272 for Case 1, which is significantly higher than the nonsingular token cases. Thus, for ALL the singular tokens where the manifold hypothesis is rejected for small radii, when the model is prompted with them, its next token probability distribution is much closer to being uniform as compared to other cases where prompts contain nonsingular tokens (or singular tokens where the manifold hypothesis is rejected for large radii).
>
> We stress that this is a quick turnaround exploratory data analysis on these points. We could do this analysis for all LLMs in our draft, but we would prefer this to be future work so as to be able to design the experiments with more intentionality. We will leverage the advice from Reviewer TdY4 on embeddings studied in control settings that will allow us to calibrate these experiments. We don’t really think that there are any trivial/easy experiments since there are many levels of confounders that exist.

---

### Official Review · Reviewer_MDro · 2025-07-03

**Clarity:** 2
**Significance:** 3
**Originality:** 3
**Rating:** 3
**Confidence:** 3

**Summary:**

This paper focuses on the structure of the token embedding space used as input for Large Language Models (LLMs). It demonstrates that this space does not possess the smooth structure typically assumed by the manifold hypothesis or the more general fiber bundle hypothesis.

The authors propose a novel statistical test to detect irregularities in the neighborhood of specific tokens and report that such irregularities are frequently observed for many tokens.

Furthermore, the paper emphasizes that its primary contribution is a "methodology for hypothesis testing on embedding points."

**Questions:**

Much of language data often possesses semantic hierarchy within its context, and understanding how this hierarchy is represented in latent space is considered a crucial research topic. I believe Robinson+2025 is one significant prior work in this area; would it be possible to conduct an analysis and discussion regarding hierarchy similar to what is done there?

**Ethical Concerns:**

["NO or VERY MINOR ethics concerns only"]

**Final Justification:**

Based on discussions with reviewers, we determined that the potential significance is high and raised the Significance rank by one level.
There is no change to the overall assessment.

**Limitations:**

The actual code and data are not publicly available. This makes it impossible to quickly verify the effectiveness of this method.

**Paper Formatting Concerns:**

The caption for Table 1 and its explanation in the main text are unclear. A more easily understandable explanation of what each numerical value in Table 1 signifies is needed.

NeurIPS guidelines encourage providing a "short proof sketch for intuition" in the main paper when proofs are included in the supplementary material.

**Quality:**

2

**Strengths And Weaknesses:**

Strength

This paper proposes a new statistical test, the "fiber bundle hypothesis," to elucidate the structure of the token embedding space in Large Language Models (LLMs).

It offers the direct test of whether token subspaces are manifolds, a topic for which prior research had not presented rigorous hypothesis testing.

The authors conducted tests on four different open-source LLMs and found "strong evidence that token subspaces are not manifolds." They also reported results indicating that tokens exhibiting irregularities often include word beginnings, word fragments, non-printable characters, and whitespace characters.


Weakness

As the authors state, if the main point of this paper is the proposal of a "methodology for hypothesis testing on embedding points," then in addition to the proofs of Theorems 1 and 2, experimental validation is needed to demonstrate the validity of this method.
Although experimental results using the proposed method are shown, there is no verification of whether these results are "valid." A comparative evaluation with results from other analysis methods is necessary. Alternatively, analysis using, for example, synthetic data where the ground truth is known, is also an option.

While rejecting regularity does imply singularity, it does not afford us any evidence as to the type of singularity.

---

> ### Author Rebuttal · Authors · 2025-07-31
>
> Thank you for your review and making us think about a topic we had yet to consider...semantic heirarchy.
>
> NB: Action items for the authors are marked with a "+" in the margin!
>
> For our rebuttal, here's a good starting point: For many tokens, we can estimate a dimension, which is at least understandable because manifolds have a well-defined dimension.  But for a few tokens--those that cause the hypotheses above to be violated--the dimension at that token is not well-defined at all!
>
> Responses to your specific concerns are below:
>
> > As the authors state, if the main point of this paper is the proposal of a "methodology for hypothesis testing on embedding points," then in addition to the proofs of Theorems 1 and 2, experimental validation is needed to demonstrate the validity .... Alternatively, analysis using, for example, synthetic data where the ground truth is known, is also an option.
>
> Your request for a visual schematic of what our test does is a great idea!  Various schematic figures were included in an earlier draft of the paper, where we had included "toy" examples of our test being run on "noisy" manifolds and stratified manifolds.
>
> However, we had to cut these examples out of the submission due to space constraints...
>
> +we will include these examples in our final version!
>
> Since we are not permitted to include graphics in our rebuttals, for the purpose of visualization, one example we plan to include is a "lollipop": a candy ball attached on the end of a short stick.  This shape has EXACTLY two important singular points.  On the interior of the stick, the space is a 1-dimensional manifold.  At the very end of the stick, though, is a singular point (the boundary).  Where the stick attaches to the "candy" part is another singularity; it is a transition from the 1-dimensional stick to the 3-dimensional candy part.  This visual example was not too difficult to construct, so your suggestion of including pictures showing cusp points, pinch points, etc. is easily implemented.
>
> > While rejecting regularity does imply singularity, it does not afford us any evidence as to the type of singularity.
>
> Correct. This is an entire branch of mathematics and a complete solution to your question would be a breakthrough of tremendous significance (Maryam Mirzakhani and Heisuke Hironaka got Fields Medals for proving partial results in this direction!).  Given the current state of this branch of mathematics, we can offer intuitive explanations for two types of singularities.
>
> (Check these out on any graphing software!)
>
> 1. A "cusp point" is a sharp point. It looks like the graph of the function $y^3 + x^2 = 0$ near the origin;
> 2. A "pinch point" looks like the graph of $z^{2}+x^{2}-y^{2}=0$.
>
> > Much of language data often possesses semantic hierarchy within its context, and understanding how this hierarchy is represented in latent space is considered a crucial research topic. I believe Robinson+2025 is one significant prior work in this area; would it be possible to conduct an analysis and discussion regarding hierarchy similar to what is done there?
>
> Could you please clarify which reference you are referring to via "Robinson+2025"?  The paper with that citation in our present submission -- [Robinson et al, Probing the topology of the space of tokens with structured prompts, arxiv:2503.15421, 2025] -- does not establish anything hierarchical according to our interpretation.  If it does in your interpretation, we would request you to clarify how.
>
> The fact that ontologists organize language into semantic hierarchies suggests that the idea of embedding tokens into a single, common space, might not be the most effective representation.  In that vein, it is entirely possible that the appearance of singularities is an indication that the idea of using an embedding representation is ill-conceived.
>
> While an intriguing possibility, an alternative token representation has not been discovered in the literature.
>
> Here is another line of thinking, inspired by [Jakubowski et al, Topology of Word Embeddings: Singularities Reflect Polysemy, in Proceedings of the Ninth Joint Conference on Lexical and Computational Semantics, 2020].  Instead of an hierarchy, it's a link of sausages of many different dimensions. Let's take the * symbol, which happens to be singular for Mistral 7B (See Table 7, number 29). In natural language, it has a very distinctive meaning in terms of calling out footnotes, or marking text as bold in Markdown, but in programming it has mathematical meaning as multiplication.  Therefore, this character connects these different senses of natural language with programming languages.  Suppose that each domain/language were in fact well-described by a manifold.  Then they would be joined/glued together in the entire latent space at the * which with high likelihood would be a singularity because the dimensions for different langauges are different [Tulchinskii et al, Intrinsic Dimension Estimation for Robust Detection of AI-Generated Texts, arxiv:2306.04723, 2023].
>
> So there are two ways our test has shown a violation of the manifold hypothesis. The large distribution of local dimension (as seen in the histogram in Fig 2 and Table 1) rejects that the token embedding is globally a manifold. This does not preclude that the embedding space is a disjoint union of smooth manifolds of different dimensions (which would still be problematic since LLMs leverage mathematical machinery such as inner products). Our test also rejects this possibility because we find tokens where a local dimension is simply not well defined. This implies something deeper like pinch points that link spaces of varied dimension. So in the lollipop example, the link point between the candy and the stick is singular because if we are at that point and look left, we appear to be in 1D while if we look right, we appear to be in 3D. The inner product that is being leveraged is on the extrinsic space and not the actual embedding space, which implies that the model is not operating on the token we would have wanted/expected. In fact, an inner product is not defined at this point (at least not in the usual way) and so we would have to characterize the singular point and find the correct mathematical machinery (if it exists).
>
> > The actual code and data are not publicly available. This makes it impossible to quickly verify the effectiveness of this method.
>
> +We will make the code publicly available once released by our government sponsor. There have been some delays due to unforeseen events; however, the sponsor did clear the present submission for release, so we do not anticipate there to be any issues with clearing the code release eventually.
>
> > The caption for Table 1 and its explanation in the main text are unclear. A more easily understandable explanation of what each numerical value in Table 1 signifies is needed.
>
> Yes, there is a lot of content in that table.
>
> +In the revision, we can certainly explain it in detail.  The following will be inserted into Section 3.3, probably around line 225.
>
> First of all, there are two main tests we run on the data (See lines 62-26): one test against manifolds, and two versions of a test for a fiber bundle
>
> The "rejects" columns give (top rows of each cell) the number of tokens that fail our test.  This is also the number of rows in Tables 3-9.  The listed p value is the smallest such p-value, which is the overall p-value for the whole test.
>
> The "dim." column shows the quartiles for the estimates of dimensions, at least for the tokens for which a dimension can be reliably be ascribed.  Note that the whole point of our test is that for each of the tokens listed in Tables 3-9, **no dimension estimate is possible**.  Quartiles are a nice way to summarize a distribution.  Usually
>
> - Q1 = 25-th percentile, so 25% of the dimensions estimated are below this quantity
> - Q2 = 50-th percentile, i.e. the median
> - Q3 = 75-th percentile, so 25% of the dimensions estimated are above this quantity
>
> > NeurIPS guidelines encourage providing a "short proof sketch for intuition" in the main paper when proofs are included in the supplementary material.
>
> We will provide a proof sketch for both Theorems in the main body, in Section 3.1 and Section 3.2.
>
> +A sketch for Theorem 1:
>
> If the assumed space is a manifold of dimension $d$, then the volume $v$ of a ball of radius $r$ centered at any point is given by an equation of the form
> $$
> v = K r^d + (\text{correction terms})
> $$
> where $K>0$ is some constant (involving $\pi$ and $d$), and the correction terms involve curvature and reach $\tau$.  Taking the logarithm of both sides yields a linear regression problem for $d$
> $$
> \log v = K + d \log r + (\text{correction terms})
> $$
> where the dimension $d$ forms the slope of the curve.
>
> For the more general case of a fiber bundle, the equations above must be modified to handle the case where the ball of radius $r$ "sticks out of" the space, and so the slope of the curve must decrease in that case.
>
> +Theorem 2's proof is already quite short, so we might (depending on space constraints) include it in full, or simply state that it is a matter of simply satisfying the bounds from [Robinson et al., 2025].

---

> > ### Comment · Reviewer_MDro · 2025-08-04
> >
> > Thank you for your kind and thorough responses.
> > I now have a clear understanding of the authors' main points.
> >
> > Regarding my question about Robinson+2025, it was based on a misunderstanding on my part, and I would like to withdraw it.
> > I apologize for any confusion this may have caused.

---

> ### Author Response · Authors · 2025-08-08
>
> Understood, no problem.
>
> Thank you once again for your review. We are happy that we were able to answer your questions satisfactorily and provide clearer understanding.
>
> Additionally, please note that in response to some reviewers' comments about the number of singular tokens, we attempted a quick exploratory data analysis of singular tokens relative to the other tokens. In case you are interested to see the results, they are in our responses (i.e. official comments) to Reviewers ZHkg and XXVi.

---

> > ### Comment · Reviewer_MDro · 2025-08-09
> >
> > I apologize for submitting my comments so close to the deadline for discussion.
> > I consider this paper to be a proposal for a new methodology based on the results of Theorem 1.
> >
> > However, as I do not have sufficient knowledge of differential geometry, I will reserve judgment on the validity of this theorem and its proof and leave it to the other reviewers and the area chair.
> >
> > I read the discussions with other reviewers with great interest. I consider that the rejection of the manifold hypothesis (detection of candidate singular points) using the statistical test method proposed in this paper may provide new insights into the mechanisms of language processing using transformers and other techniques.
> >
> > However, I believe it would be beneficial to provide concrete examples or experimental results that suggest this potential. What are the advantages of identifying candidate singularities? I believe there are various possible applications. If experimental results confirming at least one of these applications are presented, it would help to reinforce confidence in the future potential of the methodology proposed in this paper.

---

> > > ### Author Response · Authors · 2025-08-09
> > > **Better late than never...**
> > >
> > > As we have always intended this to be a methods paper, the concrete example you are seeking is not a trivial task. We have conducted some very high-level EDA (response to Reviewers ZHkg and XXVi), and we can speculate on that analysis further down.
> > >
> > > Again, maybe a generic analogy might be beneficial here as well. As Reviewer ZHkg has pointed out, there are methods that exist to determine regions of regularity (even when they are of different dimensions). If you are a surgical oncologist, knowing where the tumor is located is necessary. Being able to determine between the boundaries of the healthy vs cancerous tissue is your bread and butter. It’s the first step either towards resolution or mitigation.
> > >
> > > One resolution might be to investigate the singularities and realize that the number for our token alphabet is wrong. We found one token dictionary to have 768 tokens and discovered the reason is because that is the corresponding number of pixels in their graphics card, so the choice was for algorithmic and not semantic reasons. Perhaps there is a better choice? 50k also seems convenient rather than principled. This implies that the number of tokens is a hyper-parameter to be tuned (we are not aware of any research that assumes this). Could the performance of LLMs be improved by this type of effort?
> > >
> > > Personally, we think that the token embeddings will always be singular (likely stratified) due to natural language quirks. We have already referenced early work that correlates polysemy/homonyms as a potential reason for singular tokens (and provided an example in our response to Reviewer TdY4 for the * token), and we might be able to extend this example to contronyms / auto-antonyms. These are words that have opposite meaning, that is a word that is itself its own antonym (dusting, cleave, sanction). Suppose that each word is a token, then two semantically opposite objects will map to a single point. With apologies to Einstein, this creates a wormhole in semantic space where semantically distant neighborhoods are now arbitrarily close. Context should resolve this warping of semantic spaces, and so how well does attention/transformer produce resolution? If it cannot, all bets are off.
> > >
> > > In reality, tokens are not in 1:1 correspondence with words. Not only do parts of contronyms map onto a single token, but tokenization is a many to one map across the board. How well context / attention / transformers can unfurl these mappings is a big part of the mystery of how these models work. Identifying and analyzing the irregularities may help us understand the approximate limits of attention/transformers architectures.
> > >
> > > As mentioned earlier, we did conduct some exploratory data analysis where we looked at some of the structures for singular tokens vs the regular tokens. One structure is the max probability for the next token. For the singular tokens, this max probability is very small (around 0.13) whereas the regular tokens’ max probability is close to 1. One interpretation is if we view the tokens as nodes in a graph, the singular tokens have higher connectivity (because the probability is small and close to 0, it will choose a wider variety of next tokens almost uniformly) relative to the regular token which will have much lower connectivity (because it will only choose a consistent next token with high probability). This is analogous to the hub and spoke network models where hubs are highly connected nodes and therefore much more consequential for the network. This experiment only looks at a single token and the probability of its next token (and we reiterate that this experiment was done in the last 72 hours so it is likely sub-optimal); setting this up for a sequence of tokens while tracking conditional probabilities is much more complex. To understand sequences of tokens (i.e. real prompts), we first need to run experiments in controlled settings so we can calibrate our understanding (and code) before we test it in situ. Identifying the singular points gives us a set of “anchor tokens” to formulate subsequent hypothesis tests and ground the future experiments. There are lots of exploratory avenues…but designing the experiment necessitates access to metadata (corpus, architecture, hyper-parameter settings, etc.) so as to mitigate confounding effects. Some of these experiments might require private sector collaborations not only for metadata access but also for compute resources depending on the scale.
> > >
> > > This is hard on many levels, but we agree with your observation that any concrete experimental result would be beneficial. We are still working towards this goal.

---

### Official Review · Reviewer_MkzF · 2025-07-03

**Clarity:** 3
**Significance:** 4
**Originality:** 3
**Rating:** 5
**Confidence:** 4

**Summary:**

Motivated by the widespread but implicit assumption that token embeddings in large language models (LLMs) lie on a smooth manifold, this paper investigates whether that assumption holds in practice. It introduces a statistical testing framework to detect geometric irregularities in the embedding space, based on the notion of local neighborhoods deviating from smooth fiber bundle structures.

The main methodological contribution is a local hypothesis testing procedure that assesses whether individual token neighborhoods exhibit manifold-like behavior. The authors demonstrate that many tokens reside in locally irregular regions, that is, their neighborhoods violate the smoothness and flatness assumptions central to the manifold hypothesis.

The authors show that these irregularities are not mitigated by context window: even when used within context, such tokens introduce instability in model behavior. As a result, semantically similar queries may yield significantly different outputs depending on subtle syntactic differences, with potentially serious implications for robustness and interpretability in LLMs.

The authors test their methodological/theoretical approach on four open-source models: GPT‑2, Llemma7B, Mistral‑7B, and Pythia‑6.9B.

**Questions:**

# Major Questions/Comments

I will consider improving the clarity score and overall score if the following points are addressed:

---

## 1. Structure

**a)** Section 1.1: Please enumerate the contributions (statistical test, context, propagation, etc.) with reference to the specific parts of the text.

**b)** Please consider restructuring Section 2. As mentioned in the "Weaknesses" section, it is a bit difficult to start discussing the methods and theorems, which are said to be explained in the next section, even though they have only been briefly introduced by that point (e.g., in Figure 2), and lack accompanying explanation. Theorem 1 is referenced but not yet formulated until the following section. One suggestion is to merge Sections 2 and 3.1, but this is not the only possibility.

**c)** Please consider adding a subsection after line 184, as this is where you begin discussing singularity propagation through the transformer.

---

## 2. Mathematical formalism

I am aware that it is a non-trivial task to align a notion of a *“minimal but sufficient set of definitions”* with the expectations of the reader. That said, I have listed several terms that I believe are important to define explicitly. Terms such as *embedded reach*, *curvature*, *volume form*, *radius*, *pushforward*, *homeomorphism*, and *diffeomorphism* are used without explanation.

Additionally, it would be helpful to formally define *cusp point* and *pinch point* (unless they are supposed to be used informally), in which case I would appreciate a brief explanation.

---

## 3. Hypothesis test formulation

While the testing framework is described, clarity would be improved by explicitly stating the hypotheses in standard form:

- Null hypothesis (H₀): …
- Alternative hypothesis (H₁): …

Ideally, this should be added at the end of Section 3.1, just before the discussion on limitations of the contextual window to solve irregularity. It would also be beneficial to clearly formulate the computed statistics (you may use additional algorithmic clauses if needed). The main point is to provide a clear and concise test description.

---

## 4. Visual illustration design

Schematic diagrams or 3D (compared to 2D) illustrations of manifolds and fiber bundles (with small and large radii) could significantly enhance material accessibility and clarity. Within the same figure, you could also visually explain concepts such as *base space*, *radius*, *volume*, *cusp point*, *pinch point*, etc.

This would help readers build intuition, which can be leveraged when interpreting Figure 1. Additionally, this visual could potentially replace Figure 3a (left). While I acknowledge the authors’ effort in the current manuscript, I personally found the visuals somewhat difficult to interpret and believe they reduce overall clarity.

---

## 5. Results interpretation

Some results and conclusions would benefit from more detailed explanations and clarification:

**a)** Line 236: How were these general trends obtained? Were they based on manual inspection? Please elaborate and consider updating the manuscript.

**b)** Lines 200–306: Similarly, how were these results derived? Were they also from manual inspection? Please clarify and consider updating the manuscript.

---

## 6. Methodological clarity

The authors introduce a new test that has no existing history of successful application in prior work and apply it directly to real-world data (i.e., the embedding space of open-source LLMs), effectively without rigorous control over the data.

Would it be better to include a small experimental section using synthetic manifolds or fiber bundles, where the underlying structure is known by controlled generation? This would allow you to validate the test in a controlled setting and make your claims more convincing. I would like authors to consider incorporating such experiments, or to know their opinion on why they might disagree.



# Minor Questions/Comments

Questions/comments that are minor compared to the previous set, but I encourage the authors to consider them as they can further improve the quality of the paper:

1. Line 12: “Failure to reject null hypothesis…” is a repetition of a very similar phrase above.

2. Line 29: I wonder if this statement can be supported by prior work. While it intuitively seems to be true, it would be interesting to know if anyone has studied and quantified it.

3. Line 59: Define the neighborhood  $B(\psi)$.

4. Line 71: Should it be “global token embedding space” instead of “global token embedding”? Otherwise, please ensure that the term *embedding* is not used interchangeably to mean a single instance, an embedding function, or an embedding space.

5. Figure 1: Is there any particular reason why t-SNE is used for the large-scale view and PCA for the fine-grained structure? Why not build a three-dimensional embedding with either t-SNE or PCA and then zoom in? I could not find clarification.

6. Line 99: “In our test case, the basic assumption…” — I agree that LLMs operate in a token subspace rather than the full latent space. However, what *test case* is referred to, and why is this assumption made? I believe this is just perhaps inaccurate phrasing, so please resolve the confusion.

7. Figure 1 reference: The first reference to Figure 1 appears in the text at Line 215 (three pages later). Consider either referencing it earlier or moving the figure closer to where it is discussed.

8. Figure 2: Why does the pipeline at the top not differentiate between small and large radii?

9. Figure 3b: Should the label be “Reject fiber bundle (small radius)” instead of “Reject manifold (small radius)” for the first slope change in the green curve?

10. Figure 3b: The blue curve appears to more closely resemble the top image in Figure 3a, which would suggest rejecting the fiber bundle as the slope increases. Why is that? Please make the figures coherent or provide an explanation.

11. Figure 3a (left side): This part of the figure is a bit counterintuitive. Please consider improving the visualization (e.g., as suggested in pt. 4. Visual Illustration design in Major Comments section above) to better show the fiber bundle and connect it to the slope change dynamics in volume/radius. Additionally, even in this 2D depiction, $r_3$ seems off — it's unclear what is being represented.

12. Line 142: There is ambiguity in the notation/terminology. Is $T$ the embedding or the embedding subspace?

13. Line 145: Am I correct that $\psi$ is used solely as an element of $\mathbb{R}^l$, and not in relation to $T$? Otherwise, please resolve the confusion.

14. Theorem 1: There is a bracket inconsistency in the LHS (induced volume of $B_r(\psi)$).

15. Notation $T^w$: Please briefly introduce this notation. You use $T^{w-1}$ first, and only later define it as a sequence of tokens.

16. Line 201: Please check whether the reference is correct. Section 4.1 is the “Limitations” section, which seems like a mistake.

17. Line 218: What does “chain of sausages” mean?

18. Lines 221–224: Consider moving this part to a footnote.

19. Table 1: Could you please elaborate on why all quartiles for LLaMA-7B are exactly 4096? Is there an explanation for these results?

20. Line 257: It is unclear what is meant by “dependency between large and small scale variability.”

21. Line 270: It would be helpful to have a formal definition of *singularity* if this is a specific type of irregularity. In Line 51, it is stated that the token subspace in LLMs is “singular,” meaning not a manifold. However, the term “singular” and its variants are later used without clarification.

22. Line 323: Would it be possible to at least list possible *post hoc methods* and discuss the implications of determining that?

**Ethical Concerns:**

["NO or VERY MINOR ethics concerns only"]

**Final Justification:**

The authors diligently addressed all of my concerns during both the main rebuttal and discussion phases. I am satisfied with their responses and have accordingly increased my clarity score and overall rating.

**Limitations:**

Yes

**Paper Formatting Concerns:**

I did not find any formatting issues.

**Quality:**

3

**Strengths And Weaknesses:**

**Strengths**

From my perspective, the paper has the following strengths:

- **Significance:** The paper presents a well-defined motivation, namely, irregularities in the embedding space can lead to unstable model responses, which is undesirable. This justifies the need for statistical testing to identify tokens that lead to such unstable responses, especially given the current scale of LLM usage.

- **Originality:** To the best of my knowledge, the paper offers an original perspective on the problem (subject to the presence of prior work and the distinctions discussed by the authors in Section 1.2). In particular, I appreciate the authors’ effort to emphasize the distinction of their approach from dimensionality reduction techniques, which could otherwise raise questions about overlap.

- **Soundness:** The authors take a reasonable and systematic approach by examining the structural properties of the embedding space in practice. The experiments are logically structured, intuitive, and well-designed to support the methodological results of the paper.

**Weaknesses**

Below, I provide a high-level summary of weaknesses. Please refer to the “Question” section for a detailed list of comments, where I distinguish between major and minor questions/comments.

While somewhat subjective, I found that the paper occasionally **lacks clarity**, largely due to its narrative structure and presentation style:

- **Mathematical formalism**:  I appreciate the formal approach taken in the theoretical sections. However, the paper would benefit from more consistent and clearly defined mathematical terminology. Some readers may not have sufficient background in topology, differential geometry, or related areas. Given space constraints, some definitions could be moved to an appendix with appropriate references from the main text.

- **Structure**:  The structure of the paper is somewhat convoluted. There are abrupt transitions and forward references without sufficient contextual introduction, which makes the logical flow harder to follow.

- **Hypothesis test formulation**:  Although the main methodological contribution is a statistical test, it is not clearly or explicitly formulated. A standard hypothesis testing format, when authors are stating the null hypothesis (H₀) and alternative hypothesis (H₁), would significantly improve clarity.

- **Visual representation**:  While I appreciate the theoretical focus, the paper would benefit from additional and/or clearer/more intuitive visualizations. For instance, schematic or 3D representations of manifolds and fiber bundles (with both small and large radii) could enhance understanding. The existing figures are a step in this direction, but they remain difficult to interpret.

- **Results interpretation**:  Some of the results and conclusions could be explained in greater detail. Certain claims would benefit from additional justification or clarification.

I would like to emphasize that these weaknesses do not indicate any fundamental flaws. It is evident that the authors have already considered many of these points in preparing the submission. Please find my precise questions/comments/requests (both major and minor) in the section “Questions.”

---

> ### Author Rebuttal · Authors · 2025-07-31
>
> Thank you very much for the extremely detailed review!  We should be able to implement each of your suggestions without any trouble at all, as detailed below.
>
> We agree that the exposition could have been more organized.  We were struggling with the page limit and hastily moved some of the explanatory material to the supplement.
>
> NB: Action items for the authors are marked with a "+" in the margin!
>
> +We will edit Sec 1.1 as follows:
>
> Manifold test: A 2-sided hypothesis test where the null is defined as
> H_0: The token $\psi$ is sampled from a manifold embedded within $\mathbb{R}^\ell$ with reach $\tau$.
> and the alternative is defined as
> H_1: The token $\psi$ is not sampled from such a manifold (ie. we reject the null if there is any change in the slope of the log volume v. log radius curve)
>
> Two runs for the fiber bundle test: a 1-sided test for the following hypothesis:
> H0': The token $\psi$ is sampled from a fibered manifold with boundary that is embedded within $\mathbb{R}^\ell$ with reach $\tau$.
> H1': The tokens are not such a sample (ie. we reject the null if there is a slope INCREASE in the slope of the log volume v. log radius curve)
> We performed the volume v. radius estimation twice: once for larger radii, and once for smaller radii.
>
> For almost all the tokens, we can estimate a dimension.  But for the remaining tokens--those that cause the hypotheses above to be violated--the dimension at that token is not well-defined at all!
>
> Your request for a visual schematic of what our test does is a great idea!  Various schematic figures were included in an earlier draft of the paper, where we had included "toy" examples of our test being run on "noisy" manifolds and stratified manifolds. However, we had to cut these examples out of the submission due to space constraints.
>
> +we will include these examples in our final version!
>
> For the purpose of visualization, think of a "lollipop".  This shape has exactly two important singular points.  On the interior of the stick, the space is a 1-dimensional manifold.  At the very end of the stick, though, is a singular point (the boundary).  Where the stick attaches to the "candy" part is another singularity; it is a transition from the 1-dimensional stick to the 3-dimensional candy part.  This visual example was not too difficult to construct, so your suggestion of including pictures showing cusp points, pinch points, etc. is easily implemented.
>
> Responses to specific points that you raise in the Major Questions/Comments section of your review:
>
> > Please enumerate the contributions ... with reference to the specific parts of the text.
>
> +Sure!
>
> > Please consider restructuring Section 2...One suggestion is to merge Sections 2 and 3.1...
>
> +Section 2 indeed needs some help, but your suggestions (along with those of the other reviewers) are very do-able.  We will also move material from A.3 and A.4 into Section 2 as well for completeness.
>
> > Please consider adding a subsection after line 184, as this is where you begin discussing singularity propagation through the transformer.
>
> +Good idea!
>
> > I am aware that it is a non-trivial task to align a notion of a “minimal but sufficient set of definitions” with the expectations of the reader.
>
> +The things that absolutely need to be defined (most in Section 1.2) are:
>
> * manifold
> * singularity (and separately "singular")
> * embedding
> * embedded reach
> * volume form
> * radius
> * fibered manifold
> * fiber bundle
> * p-value
> * curvature
> * pushforward
> * homeomorphism
> * diffeomorphism
>
> > Additionally, it would be helpful to formally define cusp point and pinch point (unless they are supposed to be used informally), in which case I would appreciate a brief explanation.
>
> We meant these informally, and in that sense, they're best understood visually.  The "toy examples" mentioned above can do this easily.
>
> +We'll mention at least the following in our revision.
>
> 1. A "cusp point" is something that looks like the graph of $y^3 + x^2 = 0$ at the origin.
> 2. A "pinch point" looks like the graph of $z^{2}+x^{2}-y^{2}=0$  (check this out on any 3d graphing software!).
>
> > Re: Fig 3: I personally found the visuals somewhat difficult to interpret and believe they reduce overall clarity.
>
> +With the "lollipop example" added, we will alter Fig 3(a) so that it is based on that example.  It will be considerably clearer
>
> > Line 236: How were these general trends obtained?
>
> They were obtained from a manual inspection of Tables 4-9, which are present in the Supplementary Material due to space constraints.
>
> > Lines 200–306: Similarly, how were these results derived? Were they also from manual inspection?
>
> Lines 200-206: This follows simply the fact that Tables 2-9 are nonempty; rejections were obtained.
> +We will mention this!
>
> Lines 207-214: Inspection of Table 1.  The fact that the rejection rates and the quartiles for the dimensions are non-overlapping.
> +We will mention this!
>
> Lines 215-224: Manual inspection.  This should be taken as suggestive.  Given that the previous paragraphs established that manifold rejections occur, we are pivoting to understand possible causes.
> +We will explain that this is less about drawing conclusions about the specific reasons and more about developing the hypotheses/intuition for future study.
>
> > Would it be better to include a small experimental section using synthetic manifolds or fiber bundles, where the underlying structure is known by controlled generation?
>
> +We did run such experiments already and can include them in the revision!  Specifically, we've run spheres, planes and tori (of various dimensions), and a handful of stratified spaces/fiber bundles.
>
> We can include a toy stratified space to elucidate singular v regular points in the paper and put the rest of the examples into the supplement (and we will include all of the code publicly).
>
> > Line 29: I wonder if this statement can be supported by prior work.
>
> We haven't found anything that says this directly, but here are a few examples that are mathematically assuming manifolds:
> 1. Inner products
> 2. Cosine Similarity (which is just inner products on n-spheres which has even more assumptions about curvature especially negative curvature)
> 3. Manifold learning (Any sort of dimensionality reduction, etc)
>
> In reality, the structure of LLMs (token in -> token out) implies that the token space itself is closed under these operations.
>
> > Line 59: Define the neighborhood
>
> +This is a subtle typo.  It should be "whether *any* neighborhood of the token contains an irregularity"
>
> > Line 71: Should it be “global token embedding space” instead of “global token embedding”?
>
> If we say anything other than "embedding function", it's a typo!  In this case, we mean the "global embedding function".
> +We will check closely!
>
> > Figure 1: Is there any particular reason why t-SNE is used for the large-scale view and PCA for the fine-grained structure?
>
> We tried various "looks" and this was the clearest.
>
> > Line 99:
>
> +Oops.  "In our test case" should be deleted.
>
> > The first reference to Figure 1 appears in the text at Line 215...
>
> +Yup. Thanks.
>
> > Figure 2: Why does the pipeline at the top not differentiate between small and large radii?
>
> +We just run the process twice.  We'll edit Fig 2 accordingly.
>
> > Figure 3b: Should the label be “Reject fiber bundle (small radius)” instead of “Reject manifold (small radius)” for the first slope change in the green curve?
>
> No.  The first slope change decreases the slope; the fiber bundle test fails if the slope *increases*.
>
> > Figure 3b: The blue curve appears to more closely resemble the top image in Figure 3a, which would suggest rejecting the fiber bundle as the slope increases. Why is that?
>
> No. The top curve on Fig 3(a) has a slope increase through the slope discontinuity, whereas the blue curve in Fig 3(b) has a slope decrease.
>
> > Line 142: There is ambiguity in the notation/terminology. Is the embedding or the embedding subspace?
>
> The issue is the grammar.  The embedding function is $e$, and the token subspace is $T$.  The function $e$ is "embedding" the set $T$ into the latent space.
> +We'll fix.
>
> > Line 145: Am I correct that $\psi$ is used solely as an element of the latent space?
>
> +Yes!
>
> > Theorem 1: There is a bracket inconsistency
>
> +Thanks!
>
> > Please briefly introduce $T^w$.
>
> +It's a sequence of tokens of length $w$
>
> > Line 201: Please check whether the reference is correct.
>
> +Good catch!
>
> > Line 218: What does “chain of sausages” mean?
>
> It's an intuitive attempt at capturing the notion of a "pinch point", i.e. two spheres glued together at the north pole of one and the south pole of the other.  See above!
>
> > Lines 221–224: Consider moving this part to a footnote.
>
> +Sure.
>
> > Table 1: Could you please elaborate on why all quartiles for LLaMA-7B are exactly 4096? Is there an explanation for these results?
>
> It appears to be due to quantization.  We don't want to speculate too much, but it looks like many of the tokens are exactly equally spaced in some parts of the token subspace, almost like they were "snapped to grid".
>
> > Line 257: It is unclear what is meant by “dependency between large and small scale variability.”
>
> Euclidean space is "regular" in the sense that translations are homeomorphisms, so then a singular space cannot exhibit this kind of self-similarity.
> +We were being a bit vague and can clarify.
>
> > Line 270: It would be helpful to have a formal definition of singularity if this is a specific type of irregularity. ...
>
> This is an *entire* branch of mathematics!!  Our test is not strong enough to make this classification. Maryam Mirzakhani and Heisuke Hironaka got Fields Medals for proving partial results in this direction!
>
> > Line 323: Would it be possible to at least list possible post hoc methods and discuss the implications of determining that?
>
> We can speculate, and are happy to do so, but given our response to your query immediately above, it's really, really hard mathematics!

---

> ### Comment · Reviewer_MkzF · 2025-08-06
>
> Dear Authors,
>
> Thank you for engaging in the discussion regarding the concerns I raised. Most of the points, both major and minor, have been addressed satisfactorily, and I trust the authors will incorporate these revisions in the next version of the manuscript.
>
> However, one important issue remains unresolved - specifically, *Major Comment 6 (Methodological Clarity)*. I have not found a response to this point, unless it was overlooked. I kindly request that you provide a detailed explanation addressing this concern.
>
> Additional action items:
>
> - For *Major Comment 5a (Interpretation of Results, Line 236)*, the discussion should explicitly reference the supplementary material. Without this, the statement appears disconnected from the main text and lacks proper context.
>
> - Regarding *Minor Comment 21*, it would be preferable to either provide appropriate citations for the definitions mentioned or clearly state that these definitions are used informally (although this approach is not ideal).
>
> - Regarding *Minor Comment 22*, similarly, it would be advisable to either add proper citations for the definitions or omit these statements altogether, as their current form introduces further confusion.

---

> > ### Author Response · Authors · 2025-08-07
> >
> > Thanks for taking the time to look over all of our rebuttal comments.
> >
> > I believe that we addressed Major Comment 6 (methodology clarity) by this portion of our rebuttal:
> >
> > ===
> >
> > Your request for a visual schematic of what our test does is a great idea! Various schematic figures were included in an earlier draft of the paper, where we had included "toy" examples of our test being run on "noisy" manifolds and stratified manifolds. However, we had to cut these examples out of the submission due to space constraints.
> >
> > +we will include these examples in our final version!
> >
> > For the purpose of visualization, think of a "lollipop". This shape has exactly two important singular points. On the interior of the stick, the space is a 1-dimensional manifold. At the very end of the stick, though, is a singular point (the boundary). Where the stick attaches to the "candy" part is another singularity; it is a transition from the 1-dimensional stick to the 3-dimensional candy part. This visual example was not too difficult to construct, so your suggestion of including pictures showing cusp points, pinch points, etc. is easily implemented.
> >
> > ===
> >
> > The lollipop would serve as a nice stratified manifold to explain our method and provide the additional clarity. We have simpler examples that we will add into the supplement (with a pointer/citation for additional explanation) if needed. For a fiber bundle example, we could include a bounded cylinder or a möbius band with finite width (but this might not be helpful for clarity).
> >
> > To address the additional 3 comments:
> >
> > 1. We will explicitly provide a citation to the supplement
> > 2. We will provide a citation(s) for definitions
> > 3. We will provide a citation and limit our verbiage to reduce confusion.
> >
> > Given the page constraints, we will likely provide more specific citations and limit our own text.

---

> > > ### Author Response · Authors · 2025-08-08
> > >
> > > Separately, please note that in response to some reviewers' comments about the number of singular tokens, we attempted a quick exploratory data analysis of singular tokens relative to the other tokens. In case you are interested to see the results, they are in our responses (i.e. official comments) to Reviewers ZHkg and XXVi.

---

> ### Comment · Reviewer_MkzF · 2025-08-08
>
> Dear Authors,
>
> Thank you for your clarification. I believe your responses (including those to the other reviewers regarding exloratory analysis) have addressed my final concerns. Accordingly, I have increased my overall rating from 4 to 5 and the clarity score from 2 to 3. I wish you the best of luck with your work, regardless of the outcome of this submission.

---

### Note · Authors · 2025-08-14

We thank the referees for the spirited discussion! We are especially gratified that the referees recognize the novelty and importance of our work.

We embarked on a novel endeavor (for us): writing a statistical methods paper based on geometric theories applied to LLMs. Given page constraints, we missed the mark on our presentation to connect these topics into a clear digestible paper. To reiterate our response to reviewer ZHkg, we must speak to a broader audience.

We will provide the necessary mathematical background and intuition, through a series of "toy" examples: a sphere (a manifold), a finite width Moebius band (a fiber bundle, not a manifold), and a "lollipop" (neither), followed by our results from the 4 LLMs.

Emphasizing our overarching message: the tokens do not embed into a low dimensional smooth manifold (Thm 1)--widely assumed in the AI/ML research community--and context cannot resolve this (Thm 2). We develop a test that not only rejects this hypothesis but also gives us high level structure: a heterogeneous dimensional space containing singular points. The heterogeneity itself is enough evidence for rejection. The singular tokens give us previously unreported structure, a novel result independent of LLM behavior.

Our findings have potential implications for explaining LLM behavior. To address the scarcity of singular tokens, we conducted EDA and found variability of these points to be significantly higher than regular points (by the eyeball test). While more principled evidence is needed, we may be able to engage the stochastic parrot hypothesis: does the existence of singular tokens provide a means to test ergodicity? Can we use singularities to quantify randomness of LLM outputs?

To reiterate, our contributions are: 1. a novel statistical test, 2. evidence to reject the manifold hypothesis for several LLMs, and 3. the identification of an otherwise unreported/novel structure in these embeddings. This is the paper we hope meets the high standards of NeurIPS. The study of the singularities for LLM token embeddings is future research work in our queue.

Given the aggregation of training data across domains (and even across natural languages) the potential for internal contradictions (contronyms) and confounders (natural language, tweets, source code), it is not surprising to see singularities in the embedding.

---

### Decision · Program_Chairs · 2025-09-17

**Decision:**

Accept (poster)

**Comment:**

This paper considers a simple question: Do token embeddings satisfy the manifold hypothesis? The manifold hypothesis is a popular explanation for the success of deep learning, particularly in vision applications. But whether or not language models exploit a similar structure is (surprisingly) unknown. The authors adopt a rigorous approach to study this problem and show that in fact, token embeddings *do not* satisfy a smooth version of the manifold hypothesis. This accomplished by formulating the problem as a hypothesis test, and developing the necessary theory to develop a method for testing this hypothesis.

Reviews were split and there was extensive discussion on this submission (almost twice as many comments as average). Reviewers all agreed that the problem is relevant and the formulation provided here is correct and appropriate. There was some disagreement on the level of formalism used, which is a matter of taste and subjectivity. Given that the manifold hypothesis invokes many mathematical nuances that may be unfamiliar to some audiences, this paper does an admirable job of traversing the landscape. In particular, the role of regularity and smoothness in formulating this hypothesis is carefully laid out for readers. At the same time, one reviewer commented that such regularity may not always be necessary, and the authors should clarify this point in the camera ready. Furthermore, during the discussion, the authors helpfully clarified several points and promised to include more exposition and discussion in the final version.

Ultimately, this paper tackles an important problem in a novel way, provides a new framework for studying it, and contributes both rigorous theory and a practical method. Experiments on real LLMs (GPT, Llama, Mistral, Pythia) demonstrate the usefulness of the results.